# Improving Reasoning Ability of Large Language Models via Iterative Uncertainty-Based Preference Optimization

## Abstract

Direct Preference Optimization (DPO) has recently emerged as an efficient and effective method for aligning large language models with human preferences. However, constructing high-quality preference datasets remains challenging, often necessitating expensive manual or powerful LM annotations. Additionally, standard DPO exhibits suboptimal performance in complex reasoning tasks, such as mathematical and code reasoning. In this paper, we introduce an approach to collect preference pairs through iterative sampling and execution feedback, tailored to the current learning state (*e.g.* well-learned, mis-learned, and unlearned) of the policy model. To alleviate the failures of DPO and improve its applicability in reasoning tasks, we propose IUPO, an iterative uncertainty-based preference optimization method that achieves fine-grained preference control by assessing model confidence. We validate our approach across three reasoning tasks, incorporating five established reasoning datasets and one self-curated dataset. Our experimental results demonstrate an overall improvement of 3.6% over the standard DPO method. Furthermore, our approach exhibits promising generalizability involving weak-to-strong (8B to 70B) and cross-model (Llama to Mistral) generalizations.

## 1 Introduction

Preference optimization has emerged as a crucial ingredient in the post-training process to advance the development of large language models (LLMs) (Christiano et al., 2017; Tunstall et al., 2023; Dubey et al., 2024). The early approaches utilize reinforcement learning (RL) to align the LLM policy with human feedback or AI-generated feedback against a reward model, denoted as RLHF (Nakano et al., 2021; Ouyang et al., 2022; OpenAI, 2023) or RLAIF (Bai et al., 2022b; Lee et al., 2024; Wang et al., 2024). To streamline this process, (Rafailov et al., 2023) proposes an offline direct preference optimization method, termed DPO, which aligns the policy directly with feedback without reward modeling. Benefiting from its simplicity and efficiency, DPO has shown impressive results in various applications, including summarization Stiennon et al. (2020), dialogue assistance Bai et al. (2022a); Anil et al. (2023), and chat benchmarks Tunstall et al. (2023).

However, in complex reasoning tasks such as code reasoning and long-chain mathematical reasoning tasks, DPO often achieves only moderate gains or even impairs performance. We conjecture that this performance gap can be primarily attributed to (1) the scarcity of high-quality preference data and (2) the limitations inherent in the alignment method for improving the complex reasoning capabilities of large language models. Specifically, while long-chain complex reasoning tasks require numerous reasoning steps to solve, most alignment data are at the instance level and cannot pinpoint specific errors in incorrect answers, thus hindering the improvement of reasoning abilities. Although some researchers explore more fine-grained preference data, such as step-level (Lai et al., 2024) preferences and preference trees (Yuan et al., 2024a), they are often costly to collect and present scalability challenges. Besides, Feng et al. (2024) points out another drawback of DPO: it can reduce the probabilities and rewards of both preferred and undesirable outputs, thereby increasing the likelihood of errors in long-chain reasoning Yuan et al. (2024a). Pal et al. (2024) further investigate the failure mode of DPO when the preferred and dispreferred outputs are minimally contrastive, finding that DPO increases the probability of the token(s) that differ, yet decreases the probability of subsequent tokens. Meanwhile, another significant area of research focuses on it-

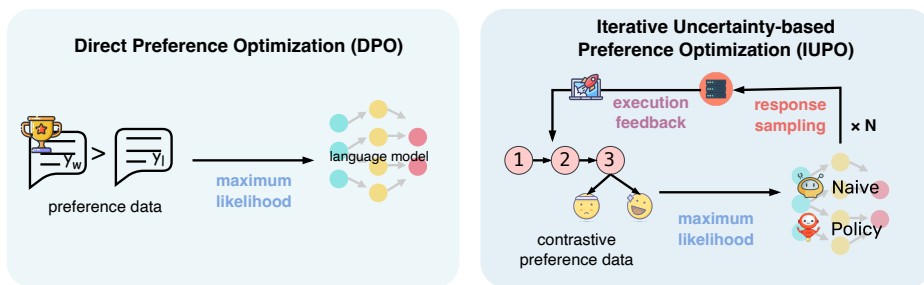

Figure 1: Comparison between DPO and our IUPO.

erative or online preference optimization, which aims to alleviate the distribution shift problem in offline DPO (Yuan et al., 2024b; Guo et al., 2024; Pang et al., 2024). However, the performance of these methods remains suboptimal, due to the challenges in ensuring the quality of preference data.

In this paper, we propose **I**terative **U**ncertainty-based **P**reference **O**ptimization (IUPO), a method that iteratively optimizes policy through response sampling and execution feedback. The overall framework of IUPO is depicted in Figure 1. Initially, our approach employs both policy and naive language models to generate multiple responses to a given query. Subsequently, we establish a virtual executable environment for the code reasoning task, and deploy answer extractors for the mathematical reasoning task, allowing us to verify the correctness of responses without reward models or verifiers. Following this, we utilize the validated data to construct preference pairs, taking into account the learning state of the policy model to improve the quality of the alignment data. The crucial advantages of the above process are outlined as follows: (1) The generation of preference data relies exclusively on pre-existing models without additional manual or more powerful model annotations. (2) The preference data is continuously updated during the iteration process, ensuring that the data remains in-distribution for policy model, which has been shown to be more effective than out-of-distribution data (Lai et al., 2024). (3) Our approach generates preference pairs with minimal contrastive (*i.e.* preferred and undesirable responses have a low edit distance), providing a better learning signal for policy optimization (D'Oosterlinck et al., 2024).

Additionally, we find the uncertainty measure (Jiang & Gupta, 2019; Wang & Zhou, 2024) strongly correlates with the performance of language models. Models tend to exhibit higher error rates when they display low confidence in certain tokens. Building on this observation, we leverage token-level uncertainty measures to achieve fine-grained control during preference optimization. Specifically, we mine the tokens that exhibit lower uncertainty measures and adjust the probability of the subsequent derailed tokens, which mitigates the decrease in the preferred probability issue. Our experimental results substantiate that the average confidence of the model is improved after optimization.

We comprehensively evaluate our method across a diverse spectrum of tasks, encompassing text-to-SQL reasoning (SQL and BIRD (Li et al., 2023)), code reasoning (Human Eval (Chen et al., 2021) and MBPP (Austin et al., 2021)), and mathematical reasoning (GSM8k (Cobbe et al., 2021a) and MATH (Hendrycks et al., 2021b)). Our experimental results demonstrate that IUPO yields a 3.6% improvement after three iterations compared to standard DPO, and consistently outperforms other baselines including SFT and DPO-Positive. In addition, our weak-to-strong and cross-model generalization experiments indicate that both our method and the generated preference data exhibit notable generalization capabilities. We also present a detailed analysis of how the uncertainty measure and iterative optimization influence the data distribution, training trajectory, and model performance.

To summarize, our key contributions are encapsulated as follows: (1) We extend the direct preference optimization methods with uncertainty measure and iterative learning, resulting in IUPO. This method endows the standard preference optimization method with fine-grained control and alleviates its distribution shift issue. (2) We introduce an automatic strategy for preference data generation through response sampling and execution feedback, which considers the learning state of the policy model without requiring additional manual or more powerful model annotations. (3) We substantiate our contributions through experimental evaluations conducted using Llama3 and Mistral models across three reasoning tasks, which conclusively demonstrate the effectiveness and generalization capability of our approach in enhancing the reasoning ability of LLMs.

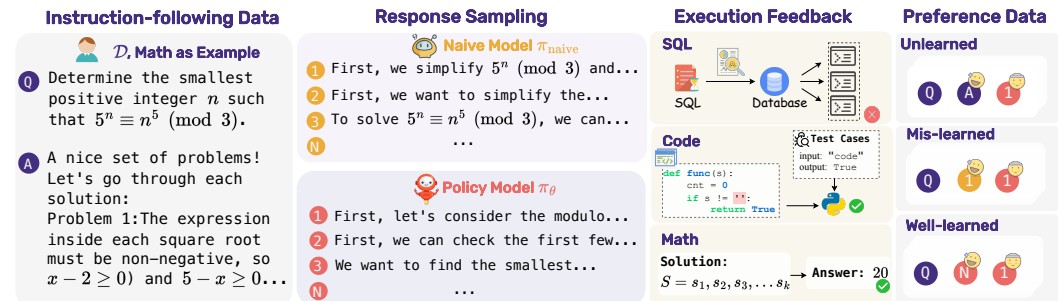

Figure 2: An illustration of the data creation pipeline.

# 2 PREFERENCE DATASET

## 2.1 DATA GENERATION

| Dataset | Prompt Length | Response Length | # Pairs | Normalized Levenshtein (↑) |
|---------|---------------|-----------------|---------|----------------------------|
| SQL | 49.5 | 280.9 | 16,627 | 87% |
| BIRD | 189.2 | 213.0 | 29,939 | 78% |
| Math | 265.5 | 1587.4 | 13,918 | 38% |
| Code | 1448.8 | 872.3 | 28,430 | 50% |

Table 1: Average character-level Levenshtein edit-distance between chosen and rejected answers for four preference datasets.

Figure 3: The number of data samples.

Traditional methods for generating high-quality preference datasets rely heavily on human labor (Ouyang et al., 2022) or strong LLMs (Bai et al., 2022b), which is time-consuming and expensive. Additionally, the precision and clarity of the resulting preference signals may be compromised, as the preference pairs are often minimally contrastive. In this section, we introduce a simple yet effective method for building preference datasets. As shown in Figure 2, our approach includes response sampling and execution feedback and can be subdivided into the following four key steps:

**Step 1: Initialization.** We begin by initializing with an instruction-following dataset $\mathcal{D}$, which consists of sets of $(x, y)$ pairs, a naive model $\pi_{\text{naive}}$, and a policy model $\pi_\theta$ initialized from $\pi_{\text{naive}}$ and then supervised fine-tuned on the dataset $\mathcal{D}$.

**Step 2: Response Sampling.** For each query $x_i$ in $\mathcal{D}$, we sample $N$ responses from both $\pi_\theta$ and $\pi_{\text{naive}}$, forming the two new set $\mathcal{D}_\theta = \{(x_i, y_j)\}_{j=1}^N$ and $\mathcal{D}_{\text{naive}} = \{(x_i, y'_j)\}_{j=1}^N$.

**Step3: Execution Feedback.** In scenarios involving code reasoning and mathematical reasoning, we simulate a virtual environment to execute synthetic responses. We then compare these execution results with the ground-truth answers to eliminate unfortunate instances. Each pair from $\mathcal{D}_\theta$ and $\mathcal{D}_{\text{naive}}$ is assigned a reward $r \in \{0, 1\}$, where $r = 1$ indicates that the response is correct.

**Step4: Preference Pairs Construction.** We construct the final preference pairs focusing on three learning states of $\pi_\theta$: (1) *Unlearned* ($y \in \mathcal{D}, y_j \in \mathcal{D}_\theta | r_j = 0$). We let the ground-truth answer as chosen and the error response generated by the policy $\pi_\theta$ as rejected, highlighting the fallibility of the model. (2) *Mis-learned* ($y'_j \in \mathcal{D}_{naive}, y_j \in \mathcal{D}_\theta | r'_j = 1, r_j = 0$). We select the correct response from the naive model as chosen to steer the deviations in the policy model. (3) *Well-learned* ($y_i \in \mathcal{D}_\theta, y_j \in \mathcal{D}_\theta | r_i = 1, r_j = 0$). In this part, we directly use the responses generated by the policy model to compose preference pairs, similar to self-rewarding (Yuan et al., 2024b).

Given that the policy $\pi_\theta$ undergoes continuous optimization during preference learning, we can naturally iterate the aforementioned steps to update preference data progressively. It is important to note that this method is not only efficient - eliminating the need for additional LMs or human labor, but also effective - it generates in-distribution preference data for the policy model. Furthermore,

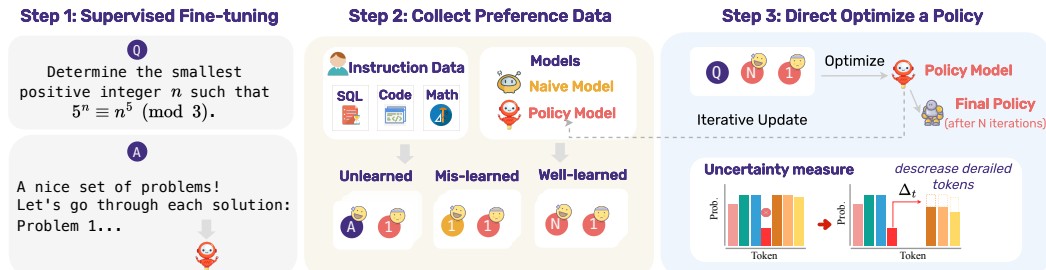

Figure 4: Overview of our IUPO framework. We first use the instruction-following data to fine-tune an LM policy. Then we collect preference data based on the learning state of the policy. Finally, we optimize the policy model with the preference data via uncertainty measure. This whole procedure is then iterated $N$ times.The circle N and circle 1 come from Figure 2.

this approach facilitates an iterative online preference optimization process. The complete algorithm process is detailed in Algorithm 1.

## 2.2 DATA STATISTICS

Regarding the instruction-following dataset, we select APPS+ (Dou et al., 2024) for code reasoning, GSM8K (Cobbe et al., 2021b) and Math (Hendrycks et al., 2021b) for mathematical reasoning, and BIRD (Li et al., 2023) for text-to-SQL reasoning. We also curated a new text-to-SQL dataset that mirrors real-world distributions. Then we apply our preference dataset generation strategy to these datasets. The statistical data and comparisons across reasoning tasks are presented in Table 1 and Figure 3. We observe that the preference pairs exhibit minimal contrast since they have low Levenshtein distance (*i.e.* edit distance), which provides more clear learning signals. For more details, please refer to Appendix A.

## 3 METHOD

### 3.1 REVISITED DIRECT PREFERENCE OPTIMIZATION (DPO)

Direct Preference Optimization (DPO) (Rafailov et al., 2023) is a computationally lightweight alignment method that directly optimizes the language model to human preferences without explicit reward modeling. Specifically, given an input prompt $x$ and a preference pair $(y_w, y_l)$, DPO aims to maximize the probability of the preferred output $y_w$ and minimize that of the undesirable output $y_l$:

$$\mathcal{L}_{\text{DPO}}(\theta) = -\mathbb{E}_{(x,y_w,y_l)\sim\mathcal{D}}[\log\sigma(\beta\log\frac{\pi_\theta(y_w|x)}{\pi_{\text{ref}}(y_w|x)} - \beta\log\frac{\pi_\theta(y_l|x)}{\pi_{\text{ref}}(y_l|x)})] \tag{1}$$

where $\mathcal{D}$ is the preference data, $\pi_\theta(\cdot)$ is the policy model to be optimized, $\pi_{\text{ref}}$ is the reference model kept frozen during training, and $\beta$ is a parameter that controls deviation from reference policy $\pi_{\text{ref}}$.

### 3.2 FAILURE MODE OF DPO

Although DPO has achieved many impressive results in various tasks and has become one of the most popular alignment methods, it only makes moderate gains or even decreases the performance on standard reasoning tasks such as code and mathematical reasoning, especially when $y_w$ and $y_l$ have low edit distance. The reasons may be attributed to the following points:

1. **Coarse-grained preference signal.** Code and mathematical reasoning are recognized as critical domains, requiring complex, long-chain reasoning abilities. However, the optimization of DPO operates at the instance level, where most preference data signals are coarse-grained, making the model struggle to identify detailed errors in incorrect answers.

2. **Decrease in preferred probability.** When preferred and undesirable responses share many similar tokens, DPO may decrease the probabilities of both the undesirable and pre-

ferred (Pal et al., 2024). Feng et al. (2024) also theoretically demonstrates DPO loss significantly impacts $\pi_\theta(y_l|x)$ due to the larger gradient, as opposed to its effect on $\pi_\theta(y_w|x)$.

3. **Frozen reward and offline learning**. Standard DPO is an offline method that relies on a pre-collected preference dataset. While the policy is continuously updated, the reward distribution remains static, leading to distribution shift and reward hacking problems.

### 3.3 IUPO

To alleviate the above failures of DPO in Section 3.2, we propose the **I**terative **U**ncertainty-based **P**reference **O**ptimization (IUPO) method, which utilize "uncertainty" to measure model confidence to achieve fine-grained control, and iterative collect preference data to optimize policy.

**Uncertainty.** Uncertainty is employed to measure model confidence (Wang & Zhou, 2024) by calculating probability disparity between the top and secondary tokens, which is similar to the minimum-margin approach (Jiang & Gupta, 2019). The uncertainty measures of a response $y$ is $\Delta(y) = \{\Delta(y^t)|t = 1, ..., |y|\}$, and $\Delta(y^t)$ refers to the uncertainty of token $t$, defined as follows:

$$\Delta(y^t) = p(y_1^t|y^{<t}, x) - p(y_2^t|y^{<t}, x) + \epsilon, \quad \Delta_t \in [\epsilon, 1] \tag{2}$$

Here $\epsilon$ is a small number to prevent the result from being zero, $y_1^t$ and $y_2^t$ represent the top two tokens at the $t$-th generation step, chosen for their maximum post-softmax probabilities from the vocabulary. Wang & Zhou (2024) utilizes the uncertainty measure $\Delta(y)$ as a reliable indicator in CoT-decoding, yielding a significant boost on the model's reasoning performance. Furthermore, our experimental observations outline the key characteristics of the uncertainty measure as follows: (1) The uncertainty measure strongly correlates with the model's performance. (2) A low uncertainty measure $\Delta_t$, indicates a lack of confidence in the model. This condition often coincides with the model's propensity to make errors, or the different tokens between preferred and undesirable. (3) The confidence of the model at the current time step has minimal impact on the generation of subsequent tokens, indicating that the model may continue along an erroneous trajectory with high confidence. Based on this, we propose integrating the uncertainty measure into the preference optimization process. On the one hand, the token-level uncertainty measure can **enable fine-grained optimization control**. On the other hand, we can leverage this measure to identify tokens where the model is prone to errors and **adjust the probability of subsequent derailed tokens** to mitigate the decrease in the preferred probability issue. Specifically, we select tokens with uncertainty measure below a fixed threshold $\tau$ and adjust the confidence of tokens within their subsequent window $K$:

$$\Delta(y^{t+k}) = (1 - \frac{k}{K}) \cdot \Delta(y^t), \quad k \in [1, K] \tag{3}$$

where $k$ is the relative distance (number of token intervals) with the token $t$, and $K$ is a hyperparameter refers to window size (number of tokens). Tokens that are closer to token $t$ within the window are more significantly influenced. Then we employ the measure to adjust the probabilities of the subsequent tokens, as illustrated in Figure 4, and the modified DPO loss can be seen as follows:

$$\mathcal{L}_{\text{UPO}}(\theta) = -\mathbb{E}_{(x,y_w,y_l)\sim\mathcal{D}}[\log\sigma(\beta\log\frac{\pi_{\theta,\Delta}(y_w|x)}{\pi_{\text{ref}}(y_w|x)} - \beta\log\frac{\pi_{\theta,\Delta}(y_l|x)}{\pi_{\text{ref}}(y_l|x)})] \tag{4}$$

where $\pi_{\theta,\Delta}(y|x) = \prod_t^{|y|} \pi_\theta(y^t|y^{<t}, x) \cdot \Delta(y^t)$, Since $\Delta(y^t)$ is less than 1, the probability of the token after the difference with preferred in $\pi_\theta(y_l|x)$ will decrease, and the corresponding gradient will be lower, thus alleviating the decrease in the preferred probability issue.

**Iterative.** To improve the performance and alleviate the reward distribution shift problem, we optimize the policy model $\pi_\theta$ iteratively, in which the policy model and the preference data are both fresh during each iteration. Specifically, we initialize the policy model $\pi_\theta$ and the reference model $\pi_{\text{ref}}$ with the supervised fine-tuned model $\pi_{\text{sft}}$. The initial preference data is also generated based on $\pi_{\text{sft}}$ and $\pi_\theta$, and is subsequently utilized to optimize the policy model $\pi_\theta$ using Equation 4. Then the preference data is regenerated based on the updated policy model $\pi_\theta$, as described in Section 2.

$$\mathcal{L}_{\text{IUPO}}(\theta) = -\sum_i^I \mathbb{E}_{(x,y_w,y_l)\sim\mathcal{D}^i}[\log\sigma(\beta\log\frac{\pi_{\theta,\Delta}^i(y_w|x)}{\pi_{\text{ref}}(y_w|x)} - \beta\log\frac{\pi_{\theta,\Delta}^i(y_l|x)}{\pi_{\text{ref}}(y_l|x)})] \tag{5}$$

where $i \in [1, I]$ is the current iteration and $I$ is the total iterations, which is set to 3 in our paper. Note that the reference model $\pi_{\text{ref}}$ kept frozen during the preference optimization process.

**Formal Analysis.** Drawing on the Equation 1, we can derive the gradient with respect to $\theta$:

$$\nabla_\theta \mathcal{L}_{\text{DPO}}(\theta) \propto -\nabla_\theta[\log \pi_\theta(y_w|x) - \log \pi_\theta(y_l|x)]$$

$$\propto -\nabla_\theta[\log \prod_{t=1}^{T} \pi_\theta(y_w^t|y_w^{<t}, x) - \log \prod_{t=1}^{T} \pi_\theta(y_l^t|y_l^{<t}, x)]$$

$$\propto -\nabla_\theta[\sum_{t=1}^{T} \log \pi_\theta(y_w^t|y_w^{<t}, x) - \sum_{t=1}^{T} \log \pi_\theta(y_l^t|y_l^{<t}, x)] \quad (6)$$

$$\propto -\sum_{t=1}^{T} \nabla_\theta[\log \pi_\theta(y_w^t|y_w^{<t}, x) - \log \pi_\theta(y_l^t|y_l^{<t}, x)]$$

where $T$ is the total number of tokens. Following Pal et al. (2024), we consider an extreme scenario that the two responses with an edit distance of 1 which differ at the token $m$ (*i.e.* $y_w = (y_1, ..., y_T)$ and $y_l = (y_1, ..., y_m', y_{m+1}, ..., y_T)$). Since the parameters $\theta$ of models are numerous, we focus on the logits $\theta_j$, which is input to softmax. We let $s_i^x$ represent the probability of the $i$-th token in the vocabulary conditioned on the input $x$, then we can simplify the Equation 6 to:

$$\nabla_{\theta_j}[\log \pi_\theta(y_w^t|y_w^{<t}, x) - \log \pi_{\theta_j}(y_l^t|y_l^{<t}, x)] = 1\{1 = j\} - s_j^{\{y_w^{<t}, x\}} - 1\{1 = j\} - s_j^{\{y_l^{<t}, x\}}$$

$$= s_j^{\{y_l^{<t}, x\}} - s_j^{\{y_w^{<t}, x\}} \quad (7)$$

Since the policy model is likely to be reasonably well optimized after SFT, we should have $s_j^{\{y_w^{<t}, x\}} \leq s_j^{\{y_l^{<t}, x\}}$ for $j \neq m$. Therefore, we see the gradient vector is increasing in the wrong logit dimensions, which shows the standard DPO may increase the probability of the incorrect token after the difference point $m$. Subsequently, the gradient of our IUPO can be derived as:

$$\nabla_\theta \mathcal{L}_{\text{UPO}}(\theta) \propto -\sum_{t=1}^{T} \nabla_\theta[\log \pi_\theta(y_w^t|y_w^{<t}, x) \cdot \Delta(y_w^t) - \log \pi_\theta(y_l^t|y_l^{<t}, x) \cdot \Delta(y_l^t)] \quad (8)$$

And the gradient of the $t$-th token with respect to the $j$-th logit becomes:

$$\nabla_{\theta_j}[\log \pi_\theta(y_w^t|y_w^{<t}, x) \cdot \Delta(y_w^t) - \log \pi_{\theta_j}(y_l^t|y_l^{<t}, x) \cdot \Delta(y_l^t)] = s_j^{\{y_l^{<t}, x\}} \cdot \Delta(y_l^t) - s_j^{\{y_w^{<t}, x\}} \cdot \Delta(y_w^t) \quad (9)$$

Since $\Delta(y_l^t) \leq \Delta(y_w^t)$ in most cases , especially when $t$ is the difference token(see Section 4.3), the gradient can be negative, thus alleviating the DPO issue described above.

## 4 EXPERIMENTS

### 4.1 EXPERIMENTAL SETTINGS

**Datasets & Baselines.** We conduct experiments using LLaMA3 series (LLaMA3-8B and LLaMA3-70B) (Dubey et al., 2024) and Mistral-7B (Jiang et al., 2023). In the supervised fine-tuning stage, we utilize the training set of BIRD (Li et al., 2023), APPS+ (Dou et al., 2024), Dart-Math (Tong et al., 2024) and self-curated text-to-SQL dataset SQL as the fine-tuning data. The preference data is iteratively generated based on the above dataset as described in Section 2. We evaluate our method on text-to-SQL reasoning tasks (SQL and BIRD), code reasoning tasks (Human Eval (Chen et al., 2021) and MBPP (Austin et al., 2021)), and mathematical reasoning tasks (GSM8K and MATH), compared with GPT series models and preference methods (DPO (Rafailov et al., 2023) and DPOP (Pal et al., 2024)). For more details, refer to the Appendix A.

**Setup.** In each preference data generation iteration, we generate $N = 10$ responses per question using sampling with temperature 0.7. In our IUPO method, we set the uncertainty threshold $\tau$ as 0.3 and the uncertainty windows $K$ as 5. All preference methods in our experiments use the same $\beta$, epoch, batch size, and learning rate, as detailed in Appendix B.

### 4.2 MAIN RESULTS

**IUPO improves over baselines.** The main performance results of all models are shown in Table 2. Across all three reasoning scenarios, it is evident that our IUPO consistently outperforms the supervised fine-tuned model (SFT) and direct preference optimization method (DPO), exhibiting an

Table 2: The main results of our IUPO against other baselines across three reasoning tasks. We report the execution accuracy in text-to-SQL reasoning, pass@1 in Code reasoning, and answer exact match accuracy in mathematical reasoning tasks. IUPO-$i$ ($i \in \{1, 2, 3\}$) refers to different iterations. **Bold** scores highlight the best performance achieved per dataset. The reported **Avg.** values are calculated by averaging performance across all datasets. We also report the performance gains and drops of our IUPO relative to the standard DPO approach.

| Model | Phase | Text-to-SQL | | Code | | Math | | Avg.↑ |
|---|---|---|---|---|---|---|---|---|
| | | SQL | BIRD | Human Eval | MBPP | GSM8K | MATH | |
| GPT-3.5-Turbo | - | 24.1 | 47.2 | 64.9 | 77.0 | 92.0 | 42.5 | 54.1 |
| GPT-4-Turbo-0409 | - | 46.4 | 53.4 | 87.6 | 80.2 | 94.5 | 73.4 | 71.1 |
| GPT-4o-0513 | - | 42.4 | 56.1 | 90.2 | 81.4 | 95.8 | 76.6 | 72.2 |
| Mistral-7B | Base | 5.2 | 27.1 | 34.2 | 47.5 | 45.9 | 16.5 | 29.4 |
| | SFT | 50.9 | 54.1 | 24.4 | 46.7 | 82.3 | 42.3 | 50.1 |
| | DPO | 49.1 | 54.2 | 23.8 | 45.9 | 83.6 | 42.3 | 49.8 |
| | DPOP | 50.0 | 54.4 | 25.0 | **47.9** | 83.2 | 42.5 | 50.5 |
| | IUPO-1 | 53.5 | 54.4 | 28.7 | 43.6 | 83.5 | 42.2 | 51.0 |
| | IUPO-2 | 54.3 | 54.7 | 29.9 | 44.2 | 83.6 | 42.5 | 51.5 |
| | IUPO-3 | **55.1** ↑6.0 | **55.1** ↑0.9 | 30.5 ↑6.7 | 44.4 ↓1.5 | 83.8 ↑0.2 | 42.8 ↑0.5 | 51.9 ↑2.1 |
| Llama3-8B | Base | 9.5 | 32.9 | 59.2 | 53.3 | 51.0 | 21.2 | 37.8 |
| | SFT | 50.0 | 52.7 | 40.9 | 57.6 | 82.5 | 43.5 | 54.5 |
| | DPO | 50.8 | 52.5 | 38.4 | 55.3 | 82.6 | 43.5 | 53.9 |
| | DPOP | 51.2 | 52.5 | 36.6 | 57.2 | 83.2 | 43.9 | 54.1 |
| | IUPO-1 | 51.7 | 54.2 | 47.6 | 56.0 | 83.2 | 43.9 | 56.1 |
| | IUPO-2 | 52.6 | 54.6 | 48.8 | 58.8 | 83.5 | 43.8 | 57.0 |
| | IUPO-3 | **52.6** ↑1.8 | **56.1** ↑3.6 | **49.0** ↑10.6 | **59.1** ↑3.8 | **83.8** ↑1.2 | **43.9** ↑0.4 | **57.4** ↑3.6 |

improvement of +2.1% in the Mistral-7B model and +3.6% in the Llama3-8B model. Additionally, we find that DPO underperforms compared to SFT across multiple datasets, particularly in scenarios where the edit distances between preferred and dispreferred examples are minimal. DPOP adds an additional penalty term to the DPO loss function to incentivize maintaining a high log-likelihood of the preferred completions. While this approach yields slightly better performance over the standard DPO, it remains less effective compared to our IUPO.

**Iterations of IUPO yield improved reasoning.** Our observations indicate that our IUPO yields performance improvements over its training iterations in most scenarios. Specifically, the average performance increases from 56.1% to 57.0% to 57.4% across each iteration. However, the magnitude of improvement diminishes with each iteration, as evidenced by the gains of 1.6%, 0.9%, and 0.4%, respectively. This trend suggests the presence of an upper limit on learning capacity across iterations, which is explored in detail in Section 4.4.

**Weak-to-Strong and cross-model generalization.** In our experiments, we deploy a Llama3-8B to synthesize the preference data in each iteration. Subsequently, we utilize the generated preference data to optimize the larger-scale Llama3-70B model and the Mistral-7B model, which features a different architecture. As shown in Table 2 and Table 3, the performance of both models has improved. This demonstrates that our method for preference dataset generation exhibits both weak-to-strong and cross-model generalization capabilities.

| Model (70B) | SQL | BIRD |
|---|---|---|
| Base | 38.6 | 43.6 |
| SFT | 62.9 | 61.7 |
| IUPO | **63.8** | **62.0** |

Table 3: The performance of Llama3-70B using the data generated by Llama3-8B.

### 4.3 ANALYSIS OF THE UNCERTAINTY

**Evolution of Model Uncertainty.** To understand the impact of uncertainty measures on model performance, we compare the uncertainty value between supervised fine-tuning and our IUPO approach. Additionally, we analyze the uncertainty values for both correct and incorrect model predictions across three distinct reasoning tasks. As shown in Figure 5, the average uncertainty measures of LLM across all four reasoning tasks are at a high level, which indicates that LLMs generally exhibit confidence in the content they generate. This observation aligns with the discussion in Section 3.3,

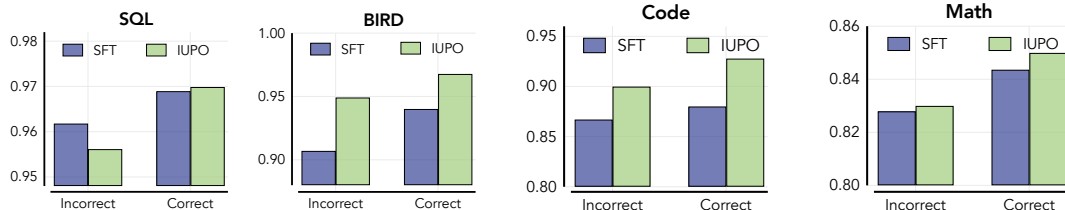

Figure 5: The uncertainty measures of SFT and IUPO between correct and incorrect answers in the four reasoning datasets. The y-axis refers to the uncertainty measure $\Delta_t$, where larger means more confidence in the model.

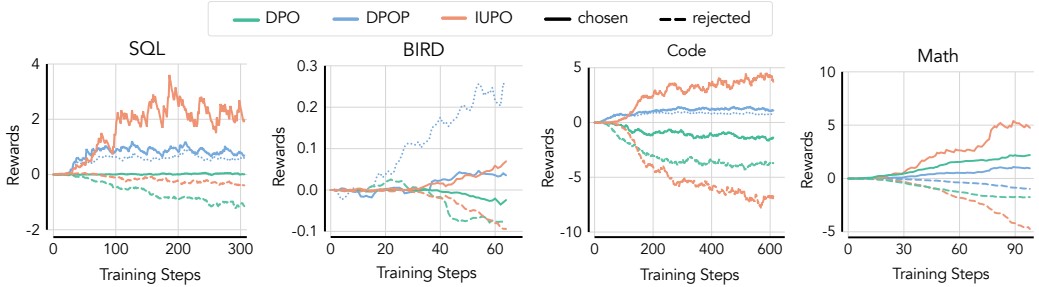

Figure 6: The reward for Llama3-8B on each reasoning task, trained using DPO, DPOP, or our IUPO alignment methods. Different methods use different colors.

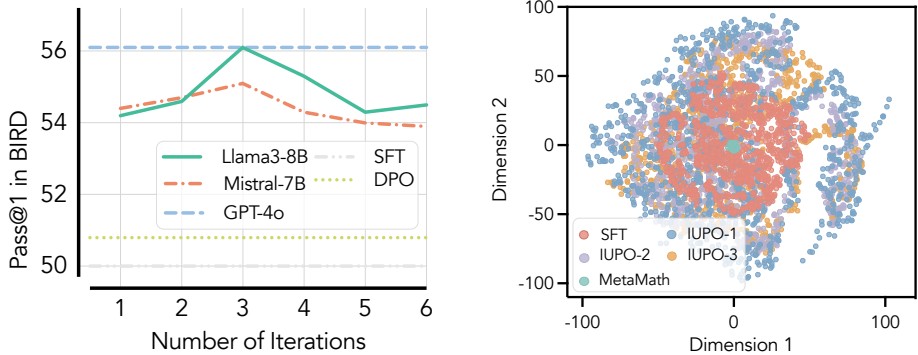

Figure 7: **Left**: Model performance for various training iterations. **Right**: Visualization of the response distribution of SFT, IUPO, and MetaMath data. We select the MetaMath data related to GSM8K and MATH for comparison.

where the uncertainty measure for the correctly predicted sample is consistently higher than that for the incorrectly predicted one. This pattern underscores the effectiveness of the uncertainty measure in identifying areas where the model is prone to making errors. Moreover, compared to supervised fine-tuning, our IUPO approach significantly boosts the confidence of the model, particularly in scenarios where the predictions are correct.

**Training Trajectory.** In Figure 6, we study the training trajectories of chosen/rejected rewards on the four reasoning tasks for DPO, DPOP, and our IUPO alignment methods. Firstly, the reward margin between the chosen and rejected of all three methods increases during the training process, indicating that these alignment methods help distinguish preferred and dispreferred responses. Secondly, the training trajectories of the three methods exhibit distinct characteristics. Specifically, DPO can reduce the reward of the chosen when the preferred and dispreferred have minimal differences. DPOP mitigates this issue but leads to an increase in the rejected rewards. In contrast, our IUPO produces a more reasonable phenomenon that the rewards of chosen grow up to positive and the rewards of rejected steadily decline. Lastly, our IUPO achieves a larger margin between preferred and dispreferred responses compared to other alignment methods within the same training steps.

## 4.4 ANALYSIS OF THE ITERATIONS

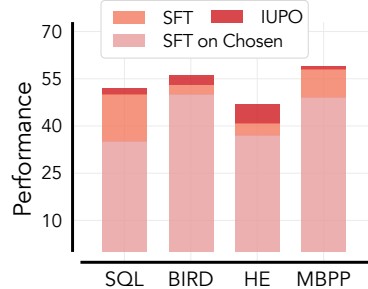

Figure 8: Comparison of SFT, SFT on chosen and IUPO.

| Phase | BIRD | Human Eval | MBPP |
|---|---|---|---|
| IUPO-1 | 54.2 | 47.6 | 56.0 |
| *w/ twice data* | 54.2 | 47.3 | 58.0 |
| *w/ triple data* | 55.1 | 47.8 | 58.8 |
| IUPO-2 | 54.6 | 48.8 | 58.8 |
| IUPO-3 | **56.1** | **47.0** | **59.1** |

Table 4: The comparative results between one iteration with more data and more iterations with updating data on Llama3-8B.

**Model performance for various iterations** To further understand the role and impact of the iterations, we visualize the relations between performance and iterations as well as the distribution of preference data for each iteration. As shown in Figure 7 Left, the performance of models on the BIRD dataset increases and then flattens out with the iterations, indicating that there is a performance ceiling when relying solely on iterative answer augmentation. To further improve performance, it may be beneficial to introduce synchronization in the diversity of questions.

**Data Distribution.** To visualize the distribution of the preference dataset, we first utilize the model to generate the pooled representation for the responses in the mathematical dataset. Afterward, we use t-SNE (Van der Maaten & Hinton, 2008) to map the representation into two-dimensional space, as shown in Figure 7 Right. The data visualized includes the supervised fine-tuning data, iterative generated preference data, and the selected data related to GSM8k and MATH from the open source MetaMath Yu et al. (2023). It is clear to find that the data from MetaMath is aggregated at the center while our iterative data broadens the boundaries of SFT data.

**Preference optimization vs. SFT on preferred.** To determine whether the performance improvements come from increased training data or the efficacy of the preference optimization algorithm, we aggregate the preferred responses curated by the model in each iteration with the supervised fine-tuning data for supervised fine-tuning. However, as shown in Figure 8, merely augmenting the dataset with preferred examples in a related manner did not help and even led to performance degradation, which is consistent with the findings in (Yuan et al., 2024b). In contrast, optimizing the model in the preference alignment manner with both preferred and dispreferred examples significantly improves the performance.

**Iterative vs. More Preference data.** We conduct a comparative experiment between one iteration with more data and more iterations with updating data to verify the effectiveness of iterative optimization. Specifically, we augment the preference data by doubling or tripling the sampling number $N$, and execute our IUPO method one iteration with the increased data. As shown in Table 4, while there is a noticeable performance improvement with the augmented preference dataset, the gains are not as substantial as performing optimization in two or three iterations. This observation underscores that the performance improvements achieved by IUPO are primarily driven by iterative optimization rather than increasing preference data volume alone.

## 4.5 ABLATION STUDY

To elucidate the individual contributions of each component within our IUPO, we conducted an ablation study, and the results are depicted in Figure 9. In this study, we systematically discard key components: the iteration process (w/o Iteration), the uncertainty measure (w/o Uncertainty, degraded to DPO), and the preference pairs that models unlearned (w/o Unlearned). The results clearly demonstrate that each component of our method produces a positive effect on performance improvement, especially the iterative optimization and the uncertainty measure.

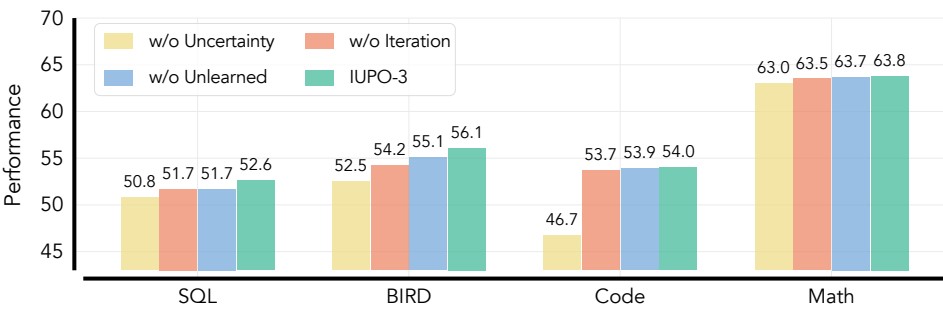

Figure 9: Ablation study across all the reasoning tasks using Llama3-8B. We show the averaged performance for the datasets of Code and Math.

## 5 RELATED WORK

### 5.1 PREFERENCE DATASET CURATION AND AUGMENTATION

Preference dataset collection is the first and important step in LLM alignment. A common preference dataset is a set of prompts paired with a preferred and dispreferred response, where the preferred embodies the instructions, intentions, preferences, and values that humans intend for the LLM to internalize and replicate. Human labeling (Christiano et al., 2017; Ouyang et al., 2022) is a crucial tool for high-quality preference dataset construction. However, it is labor intensive and necessitates a certain level of knowledge of the annotator, which increases the cost and hinders the scalability of the data scale. Recently, LLMs have shown a high degree of alignment with human judgment (Gilardi et al., 2023), some researchers focus on Reinforcement Learning from AI Feedback (RLAIF) (Bai et al., 2022b; Lee et al., 2024), which leverages strong LLMs (*e.g.* GPT-4) to generate preference labels and achieves comparable performance to human labors. In this paper, we propose an effective method to build a preference dataset via iterative sampling based on the policy model and execution feedback to verify the correctness, which is efficient and effective.

### 5.2 PREFERENCE OPTIMIZATION OF LLMS

Reinforcement Learning from Human Feedback (RLHF) (Christiano et al., 2017; Ouyang et al., 2022) has emerged as a cornerstone in aligning LLMs with human preferences, providing a mechanism to enhance LLMs' comprehension of human requirements and refining their responses for improved alignment. This approach involves training a reward model with preference data and then optimizing the policy model with the reward model. To simplify this process, Rafailov et al. (2023) proposes DPO, which directly uses the pairwise data for model optimization without reward modeling. While DPO has achieved impressive results in various scenarios, it only makes moderate gains or even decreases performance for mathematical or code reasoning. Feng et al. (2024) analyzes the failure modes of Direct Preference Optimization (DPO) and finds that the optimization process can inadvertently reduce the number of preferred examples. To alleviate this issue, Pal et al. (2024) adds a penalty term to DPO loss to incentivize maintaining a high log-likelihood of the preferred completions. In contrast, we utilize uncertainty to measure model confidence to achieve fine-grained control. Furthermore, we optimize the policy model in an iterative manner to realize online learning.

## 6 CONCLUSION

In this paper, we introduce IUPO, an iterative uncertainty-based preference optimization method via response sampling and execution feedback to improve the reasoning ability of LLMs. Our contribution also includes an automatic preference data generation strategy without additional manual or more powerful model annotations while considering the learning state of the policy model. Through comprehensive experimentation across three reasoning tasks and in-depth analysis of the components of our method, we have demonstrated the substantial benefits of IUPO in augmenting the reasoning ability of LLMs. In the future, an exciting avenue for research involves exploring IUPO in diverse datasets with more various models.

## REPRODUCIBILITY STATEMENT

The source of our self-curated SQL dataset and the preference datasets will be released soon. In order to provide support to reproduce our method and experiments, we provide the detailed source code of data generation and the implementation of DPO, DPOP, and our IUPO methods in the supplementary materials with all scripts and hyper-parameters. We provide a README script to instruct how to run the codes. We also list the details of the datasets and the hyper-parameters in Appendix.

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

## A  ADDITIONAL DATASET INFORMATION

### A.1  THE SUPERVISED FINE-TUNING DATASETS

| Dataset | # Train | Task | Source |
|---|---|---|---|
| SQL | 14,000 | Text-to-SQL | Self-Curated |
| BIRD (Li et al., 2023) | 12,751 | Text-to-SQL | Open Source |
| APPS+ (Dou et al., 2024) | 7,413 | Code Reasoning | Open Source |
| DartMath (Tong et al., 2024) | 591,000 | Math Reasoning | Open Source |

Table 5: Details about the supervised fine-tuning datasets.

Table 5 shows the statistical details of the datasets used in supervised fine-tuning phases.

**SQL Prompt**

```
Generate a SQL query to answer this question:  `{question}`

DDL statements:
{table_info}

The following SQL query best answers the question `{question}`:
```

**SQL** Since most of the prevalent Text-to-SQL benchmarks (*i.e.* WikiSQL (Zhong et al., 2017), and Spider (Yu et al., 2018)) focus on database schema with few rows of database values, we create a more challenging dataset for Text-to-SQL parsing to reduce the gap between academic study and real-world applications. In particular, we first select real-world databases with multiple rows and columns. Then we utilize GPT-4 OpenAI (2023) to generate user questions about the databases and the corresponding answers. All questions and answers are manually verified to ensure their quality.

**BIRD Prompt**

```
### Database scheme:
{table_info}

### Question:
{question}

### Match value:{match_value}

###SQL:
```

**BIRD** BIRD is another Text-to-SQL dataset developed by (Li et al., 2023). It contains 12,751 Text-to-SQL pairs and 95 databases with a total size of 33.4GB, spanning 37 professional domains, which highlights the challenges of dirty and noisy database values, external knowledge grounding, and SQL efficiency, particularly in the context of massive databases.

**APPS+ Prompt**

```
### Instruction:
write an algorithm in python:  {Task description}
### Response:
```

**APPS+** APPS+ is a clean version of APPS (Hendrycks et al., 2021a) created by (Dou et al., 2024). They excluded instances lacking input, output, and solutions of APPS, and standardized the formats of all instances. APPS+ contains 7,456 instances, including problem descriptions, canonical solutions, unit tests, and starter codes.

**DartMath Prompt**

```
Below is an instruction that describes a task.  Write a response
that appropriately completes the request.

###Instruction:
{query}

### Response:
```

**DartMath** DartMath is a synthetic dataset based on GSM8k (Cobbe et al., 2021a) and MATH Hendrycks et al. (2021b) via difficulty-aware rejection sampling Tong et al. (2024).

## A.2 DETAILS OF PREFERENCE DATASET GENERATION STRATEGY

We show the details algorithm process in Algorithm 1.

---

**Algorithm 1:** Preference Dataset Generation Strategy

---

**Input:** naive model $\pi_{\text{naive}}$, policy model $\pi_\theta$, instruction-following dataset $\mathcal{D}$ consists of $N(x, y)$ pairs, iterations $\mathcal{I}$, sampling numbers $N$

1: Initialize $\pi_\theta$ from $\pi_{\text{naive}}$
2: Preference pairs dataset $\mathcal{D}_{\text{IUPO}} = \emptyset$
3: **for** $i = 1$ to $\mathcal{I}$ **do**
4:     **for** each pair $(x, y)$ in $\mathcal{D}$ **do**
5:         $\mathcal{D}_{\text{naive}} \leftarrow \emptyset, \mathcal{D}_\theta \leftarrow \emptyset$
6:         **for** $j = 1$ to $N$ **do**
7:             $y_j = \pi_\theta(x)$ `// generate response from` $\pi_\theta$
8:             $y'_j = \pi_{\text{naive}}(x)$ `// generate response from` $\pi_{\text{naive}}$
9:             $r = \text{EX}(x, y, y_j)$ `// obtain the reward via execution feedback`
10:           $r' = \text{EX}(x, y, y'_j)$
11:            Add pair $(x, y_j, r_j)$ to $\mathcal{D}_\theta$, and add pair $(x, y'_j, r'_j)$ to $\mathcal{D}_{\text{naive}}$
12:         **end for**
13:         Add all $(x, y, y_j)$ to $\mathcal{D}_{\text{IUPO}}$ where $y_j \in \mathcal{D}_\theta$ and $r_j = 0$ `// Unleanred pairs`
14:         Add all $(x, y'_j, y_j)$ to $\mathcal{D}_{\text{IUPO}}$ where $y_j \in \mathcal{D}_\theta, y'_j \in \mathcal{D}_{\text{naive}}$ and $r'_j = 1, r_j = 0$
        `// Mislearned pairs`
15:         Add all $(x, y_i, y_j)$ to $\mathcal{D}_{\text{IUPO}}$ where $y_i, y_j \in \mathcal{D}_\theta$ and $r_i = 1, r_j = 0$ `// Well-learned pairs`
16:     **end for**
17:     $\pi_\theta = $ train_policy_model$(\pi_\theta, \mathcal{D}_{\text{IUPO}})$ `// training policy model`
18: **end for**
**Output:** $\mathcal{D}_{\text{IUPO}}, \pi_\theta$

---

### A.3 DETAILS OF THE EVALUATION DATASETS

| Dataset | # Test | Task | Source |
|---|---|---|---|
| SQL | 116 | Text-to-SQL | Self-Curated |
| BIRD (Li et al., 2023) | 1,533 | Text-to-SQL | Open Source |
| Human Eval (Chen et al., 2021) | 164 | Code Reasoning | Open Source |
| MBPP (Austin et al., 2021) | 257 | Code Reasoning | Open Source |
| GSM8k (Cobbe et al., 2021b) | 1,319 | Math Reasoning | Open Source |
| MATH (Hendrycks et al., 2021b) | 5,000 | Math Reasoning | Open Source |

Table 6: Details about the evaluation datasets.

Table 6 shows the statistical details of the evaluation datasets. We use the same data configuration and assessment as the baseline.

## B DETAILS OF EXPERIMENTS

### B.1 DETAILED EVALUATION METRICS

For all datasets, we compare the execution results between model predictions and ground truth. Specifically, we compute the **Execution Accuracy (EX)** for the Text-to-SQL reasoning task, which is defined as the proportion of examples in the evaluation set for which the executed results of both the predicted and ground-truth SQLs are identical, relative to the overall number of SQLs (Zhong et al., 2017). For code reasoning tasks, we compute the **pass@1** metric, where 1 code sample is generated per problem, and a problem is considered solved if the sample passes the unit tests. For the mathematical reasoning task, we extract the final answer from the generated solution and compare whether it is the same as the ground truth.

Table 7: Comparison results for 3 iterations of DPO, KTO, and our IUPO methods.

| Model | Phase | Text-to-SQL | | Code | | Math | | Avg.↑ |
|-------|-------|------|------|------------|------|-------|------|-------|
| | | SQL | BIRD | Human Eval | MBPP | GSM8K | MATH | |
| Llama3-8B | Base | 9.5 | 32.9 | 59.2 | 53.3 | 51.0 | 21.2 | 37.8 |
| | SFT | 50.0 | 52.7 | 40.9 | 57.6 | 82.5 | 43.5 | 54.5 |
| | DPO-1 | 50.8 | 52.5 | 38.4 | 55.3 | 82.6 | 43.5 | 53.9 |
| | DPO-2 | 48.3 | 54.3 | 37.2 | 55.3 | 82.7 | 43.3 | 53.5 |
| | DPO-3 | 49.1 | 53.0 | 37.2 | 55.6 | 83.1 | 43.2 | 53.5 |
| | KTO-1 | 49.1 | 53.0 | 36.6 | 58.8 | 82.6 | 43.1 | 53.9 |
| | KTO-2 | 48.2 | 53.7 | 37.2 | 57.6 | 82.6 | 42.6 | 53.7 |
| | KTO-3 | 52.6 | 53.7 | 37.4 | 58.4 | 83.5 | 43.4 | 54.8 |
| | IUPO-1 | 51.7 | 54.2 | 47.6 | 56.0 | 83.2 | 43.9 | 56.1 |
| | IUPO-2 | 52.6 | 54.6 | 48.8 | 58.8 | 83.5 | 43.8 | 57.0 |
| | IUPO-3 | **52.6** ↑1.8 | **56.1** ↑3.6 | **49.0** ↑10.6 | **59.1** ↑3.8 | **83.8** ↑1.2 | **43.9** ↑0.4 | **57.4** ↑3.6 |

Table 8: Comparison results of DPO and our IUPO on ARC-c and TruthfulQA.

| Dataset | SFT | DPO-1 | DPO-2 | DPO-3 | IUPO-1 | IUPO-2 | IUPO-3 |
|---------|-----|-------|-------|-------|--------|--------|--------|
| ARC-c | 60.0 | 63.5 | 64.9 | 66.8 | 63.4 | 65.2 | 66.9 |
| TruthfulQA | 52.0 | 50.0 | 52.5 | 53.8 | 51.0 | 53.4 | 54.7 |

### B.2 IMPLEMENTATION DETAILS

All models of SFT and preference optimization phases are trained in the same environment ($4 \times 40G$ A100 GPUs.) During the preference dataset generation process, we generate $N = 10$ responses per question using sampling with temperature=0.7 and topp=1.0. Due to resource limitations, we adapt LoRA training in SFT and preference optimization phases with the PEFT Mangrulkar et al., 2022 framework. We set learning rate as 1e-4, training 4 epochs in SFT and 1 epoch in the alignment process. In DPO-Positive, we set the $\lambda = 50$ same as Pal et al. (2024). In our IUPO, we set the uncertainty threshold $\tau = 0.3$ and the uncertainty windows $K = 5$.

## C ADDITIONAL EXPERIMENTS

### C.1 COMPARISON WITH PREFERENCE OPTIMIZATION METHODS

KTO Ethayarajh et al. (2024) is a powerful alignment method that directly maximizes the utility of generations instead of maximizing the log-likelihood of preferences. Since KTO does not require paired data, we split the preferred and dispreferred instances as the training data for a fair comparison. For all alignment methods, we optimize the policy model 3 iterations, and the results can be seen in Table 7. We find that KTO outperforms the vanilla DPO method and gains performance from iteration. This advantage may come from using prospect theory in KTO and its full utilization of preference samples. However, our IUPO remains the best performance due to its superior applicability to the contrastive preference data.

### C.2 EXTENSION TO MORE REASONING SCENARIOS

In some reasoning scenarios, we cannot use the execution environment to verify the correctness of the response. At this point, we can use the reward model to score the sampling responses and collect preference pairs by treating high-scoring responses as positive examples and low-scoring ones as dispreferred examples. To demonstrate the effectiveness of our method, following (Dong et al., 2024), we iteratively collect the preference pairs and optimize the policy model. Then we evaluate the model on TruthfulQA Lin et al. (2022) and ARC-c **?** datasets. TruthfulQA. Among them, TruthfulQA is a benchmark dataset to measure the truthfulness of language models, which

comprises 817 questions that span 38 categories, including health, law, finance, and politics. ARC-c is a dataset to measure the commonsense reasoning ability of models, which contains questions for science exams from grade 3 to grade 9. The results can be seen in Table 8, our IUPO obtains superior performance than DPO in each iteration, and the performance improves with iterations increase, proving that our approach is also feasible with the reward model.

# D QUALITATIVE EXAMPLES

| Question: | What are the foiled cards that are incredibly powerful when paired with non foiled cards? List the IDs. |
|---|---|
| Ground Truth: | `SELECT id FROM cards WHERE cardKingdomId IS NOT NULL AND cardKingdomFoilId IS NOT NULL` |
| SFT: | `SELECT cardkingdomfoilid FROM cards WHERE cardkingdomid IS NOT NULL AND cardkingdomfoilid IS NOT NULL` ✘ |
| DPO: | `SELECT cardkingdomfoilid FROM cards WHERE cardkingdomid IS NOT NULL AND cardkingdomfoilid IS NOT NULL` ✘ |
| IUPO: | `SELECT id FROM cards WHERE cardkingdomfoilid IS NOT NULL AND cardkingdomid IS NOT NULL` ✔ |

Table 9: A case example of BIRD dataset. ✔ refers to the answer is correct, while ✘ refers to the answer is incorrect.

| Question: | Lillian's garden doesn't have any bird feeders in it so she wants to add some. She builds 3 and buys 3 others. Each bird feeder seems to attract 20 birds throughout the day until Lillian notices that the birds seem to prefer the feeders she made herself which attract 10 more birds each than the store-bought ones. How many birds can Lillian expect to see in her garden each day if the same amount keep coming to her bird feeders? |
|---|---|
| Ground Truth: | Lillian has 3 store-bought + 3 self-made bird feeders = <<3+3=6>>6 bird feeders in all. Lillian first notices each bird feeder attracts 20 birds, 6 x 20 = <<6*20=120>>120 birds. Then Lillian notices that her 3 self-made bird feeders attract 10 more birds each, 10 x 3 = <<10*3=30>>30 extra birds. Each day Lillian can expect to see 120 + 30 = <<120+30=150>>150 birds in her garden. The answer is: 150. |
| SFT: | Lillian builds 3 bird feeders and buys 3 others, so she has a total of 3 + 3 = 6 bird feeders. Each bird feeder she made herself attracts 10 more birds than the store-bought ones, so each of her bird feeders attracts 20 + 10 = 30 birds. The total number of birds that can be seen in her garden each day is the number of bird feeders times the number of birds each feeder attracts, so it's 6 * 30 = 180 birds. The answer is: 180 ✘ |
| DPO: | Lillian builds 3 bird feeders and buys 3 others, so she has a total of 3 + 3 = 6 bird feeders. Each bird feeder she made herself attracts 20 + 10 = 30 birds each day. Each bird feeder she bought from the store attracts 20 birds each day. So, the total number of birds she can expect to see in her garden each day is 3 * 30 + 3 * 20 = 90 + 60 = 150 birds. The answer is: 150 ✔ |
| IUPO: | Lillian builds 3 bird feeders and buys 3 others, so she has a total of 3 + 3 = 6 bird feeders. Each bird feeder she made herself attracts 10 more birds than the store-bought ones, so each of her bird feeders attracts 20 + 10 = 30 birds. The 3 bird feeders she made herself can therefore attract a total of 3 * 30 = 90 birds. The 3 store-bought bird feeders can attract a total of 3 * 20 = 60 birds. Adding these together, Lillian can expect to see a total of 90 + 60 = 150 birds in her garden each day. The answer is 150. ✔ |

Table 10: A case example of MATH dataset. ✔ refers to the answer is correct, while ✘ refers to the answer is incorrect.

We show some qualitative examples of BIRD and Math tasks in Table 9 and Table 10. We compare the reasoning results of SFT, DPO, and our IUPO and highlight the wrong parts with distinct colors. In the Text-to-SQL task, we find that the results generated by different methods were remarkably similar, with only a few words differing. However, it is precisely these critical differences that lead to the opposite final outcomes. Benefiting from the contrasting preference data and fine-grained preference control, our IUPO makes correct judgments and predictions. Regarding to the mathematical reasoning task, the trajectory of logical reasoning is notably sensitive to nuanced variations, despite the apparent distinctions in formal derived from various methods. We find that the SFT model initially demonstrated correct reasoning but deviated towards incorrect conclusions in later stages. In contrast, both the DPO and UDPO models consistently made progress toward more accurate outcomes.

