# OpenReview forum: "Improving Reasoning Ability of Large Language Models via Iterative Uncertainty-based Preference Optimization"
_ICLR.cc/2025/Conference — Submitted to ICLR 2025_

### Official Review · Reviewer_D59P · 2024-11-02

**Soundness:** 3
**Presentation:** 3
**Contribution:** 2
**Rating:** 5
**Confidence:** 4

**Summary:**

The paper introduces a method called Iterative Uncertainty-based Preference Optimization (IUPO) to enhance the reasoning capabilities of large language models (LLMs). The authors address the limitations of Direct Preference Optimization (DPO), particularly its suboptimal performance in complex reasoning tasks such as mathematical and code reasoning, due to the scarcity of high-quality preference data and the limitations of its alignment method.

**Strengths:**

1. IUPO automates the generation of preference data, which is typically a labor-intensive process. It does this by leveraging existing model responses and execution feedback, eliminating the need for additional manual annotations or more powerful models.
2. IUPO employs an iterative approach to continuously update preference data, ensuring that the data remains relevant and in-distribution for the policy model. This iterative optimization is shown to be more effective than simply increasing the volume of preference data.
3. This paper conducts comprehensive experiments across three reasoning tasks using established and self-curated datasets, showing an overall improvement of 3.6% over the standard DPO method.

**Weaknesses:**

1. The application of reinforcement learning algorithms for tuning SFT models with LoRA (Low-Rank Adaptation) might be somewhat trivial, given that LoRA adds some parameters. Could you demonstrate the performance advantage of your reinforcement learning algorithm under full-parameter fine-tuning settings?
2. In terms of algorithmic performance, there are currently many algorithms that surpass the DPO(such as IPO and KTO). Relying solely on experimental validation with DPO may be somewhat insufficient and narrow. It would be advantageous to include some theoretical guidance or comparisons with other reinforcement learning algorithms to enhance the analysis.

**Questions:**

See the weakness.

---

> ### Author Response · Authors · 2024-11-25
> **Response to Reviewer D59P**
>
> Dear Reviewer 6qtw,
>
> Thank you for your valuable comments and thoughtful questions. We sincerely appreciate the time and effort you have invested in reviewing our work. We have implemented several key revisions in the draft in response to your and other reviewers’ constructive feedback. Below, we provide detailed responses to each of your comments to the best of our ability.
>
> **W1**. Thanks for your suggestions. In fact, we conduct experiments with full parameter training at first, but we find that the trend is consistent with the LoRA training. Therefore, due to the resource limitations, we chose LoRA training finally.
>
> **W2**. Since our method focuses on preference dataset construction and the failures of DPO in complex reasoning tasks, we have only compared it with a similar method DPOP. However, your comments make us realize that this is insufficient, thus we compare our method with more variant methods in Appendix C.1. The results can be also seen as follows:
> |      |      |       |  | |        |         | | | |
> |------|----------------------|------------------|---------|---------|-----------|----------|-----------|-----------|-----------|
> | Model| Phase   | SQL   | BIRD  | Human Eval   | MBPP    | GSM8K     | MATH      | Avg.     |
> | Llama3-8B | Base    | 9.5   | 32.9  | 59.2   | 53.3    | 51.0      | 21.2      | 37.8      |
> | Llama3-8B  | SFT     | 50.0  | 52.7  | 40.9   | 57.6    | 82.5      | 43.5      | 54.5      |
> | Llama3-8B| DPO-1   | 50.8  | 52.5  | 38.4   | 55.3    | 82.6      | 43.5      | 53.9      |
> | Llama3-8B| DPO-2   | 48.3  | 54.3  | 37.2   | 55.3    | 82.7      | 43.3      | 53.5      |
> | Llama3-8B | DPO-3   | 49.1  | 53.0  | 37.2   | 55.6    | 83.1      | 43.2      | 53.5      |
> | Llama3-8B| KTO-1   | 49.1  | 53.0  | 36.6   | 58.8    | 82.6      | 43.1      | 53.9      |
> | Llama3-8B | KTO-2   | 48.2  | 53.7  | 37.2   | 57.6    | 82.6      | 42.6      | 53.7      |
> | Llama3-8B| KTO-3   | 52.6  | 53.7  | 37.4   | 58.4    | 83.5      | 43.4      | 54.8      |
> | Llama3-8B | IUPO-1  | 51.7  | 54.2  | 47.6   | 56.0    | 83.2      | 43.9      | 56.1      |
> | Llama3-8B| IUPO-2  | 52.6  | 54.6  | 48.8   | 58.8    | 83.5      | 43.8      | 57.0      |
> | Llama3-8B| IUPO-3  | 52.6  | 56.1 | 49.0 | 59.1| 83.8 | 43.9 | 57.4 |
>
> We find that KTO outperforms the vanilla DPO method and gains performance from iteration but underperforms our IUPO method. We think that our approach is more suitable for these scenarios where the preferred and dispreferred examples have a small edit distance.
>
> Additionally, we want to claim that the motivations and objectives of KTO and our IUPO are different. KTO applies prospect theory to directly maximize the utility of generations instead of maximizing the log-likelihood of preferences. In the specific implementation, KTO does not require paired preference data. In contrast, we introduce an approach to collect preference pairs through iterative sampling and execution feedback, and on the other hand, we propose an iterative uncertainty-based preference optimization method that addresses the failures of DPO in complex reasoning scenarios.
>
> We highly appreciate your valuable input, and we are committed to integrating these insights into our upcoming experiments and the refined version of our work.
>
> We hope these updates adequately address your concerns and kindly encourage you to reconsider our review score in light of these clarifications. Your contribution significantly enhances the quality of our research, and we thank you for your thoughtful feedback.

---

> > ### Comment · Reviewer_D59P · 2024-11-26
> >
> > Although your results achieved the best performance in the LoRA fine-tuning scenario, using LoRA for fine-tuning inherently increases the number of the model's parameter count, so the performance improvement is expected. Additionally, based on some observations, LoRA can only modify some simple characteristics of the model (such as output formatting), and there is still a significant difference compared to full supervision fine-tuning. Therefore, I am not entirely sure if your method will definitely improve performance in the SFT (Supervised Fine-Tuning) scenario, so I will keep my score as it is.

---

> > > ### Author Response · Authors · 2024-11-28
> > > **Response to Reviewer D59P Follow-Up**
> > >
> > > Dear Reviewer D59P,
> > >
> > > Thank you for your further reply. We are very sorry that due to time and resource limitations, we are not able to add the full training of experimental results in time before.  Thanks to the extension of the review period, we further experimented on more reasoning tasks, including full training. Firstly, we add the full parameter fine-tuning experimental results on the mathematical reasoning task compared with the DPO method. The results can be seen as follows:
> > >
> > > | Model | Phase | GSM8k | MATH |
> > > | --- | --- | --- | --- |
> > > | Llama3-8B | SFT | 82.48 | 43.50 |
> > > |  Llama3-8B| DPO | 83.2 | 43.6 |
> > > | Llama3-8B | IUPO-1 | 83.62 | 44.0 |
> > > | Llama3-8B | IUPO-2 | 84.15 | 44.28 |
> > > |  Llama3-8B| IUPO-3 | 84.46 | 44.8 |
> > >
> > > It can be seen that our method still has advantages under full parameters training. Due to time constraints, we will add more results from full parameters training experimental results in the future.
> > >
> > > Secondly, we compare our IUPO and DPO in the reasoning scenarios that the execution environment is not available. Specifically, we apply a reward model to replace the oracle execution environment to score the responses sampling from policy and collect preference pairs. We use the preference pairs to optimize the policy model with either DPO or our IUPO approach. Then we sample responses from the optimized policy, score them with the reward model, and optimize the policy model iteratively. Note that all models are tuned with full parameters. The results can be seen as follows:
> > >
> > > | Metric | ARC-c | TruthfulQA |
> > > | --- | --- | --- |
> > > | SFT | 60.0 | 52.0 |
> > > | DPO-1 | 63.5 | 50.0 |
> > > | DPO-2 | 64.9 | 52.5 |
> > > | DPO-3 | 66.8 | 53.8 |
> > > | IUPO-1 | 63.4 | 51.0 |
> > > | IUPO-2 | 65.2 | 53.4 |
> > > | IUPO-3 | 66.9 | 54.7 |
> > >
> > > We can see that our IUPO obtains superior performance than DPO in each iteration, and the performance improves with iterations increase, proving that our approach is also feasible with the reward model scenarios.
> > >
> > > Thank you again for your thoughtful review and constructive feedback. We hope these additional experiments and clarifications address your concerns and kindly encourage you to reconsider your score in light of the updates! We are grateful for the opportunity to engage further.

---

> > > > ### Author Response · Authors · 2024-12-02
> > > > **Kindly Request for Re-review and Discussion**
> > > >
> > > > Dear Reviewer D59P,
> > > >
> > > > As the discussion period draws to a close very soon, we sincerely appreciate your time and expertise! We kindly ask if our new responses and revisions have addressed your concerns. If so, we respectfully request you to reconsider your assessment. Further, we are happy to address any further concerns you may have! Your valuable insights contribute significantly to the refinement of our work.
> > > >
> > > > Thank you once again for your support and collaboration.
> > > >
> > > > Best regards,
> > > >
> > > > Authors

---

### Official Review · Reviewer_yKKv · 2024-11-04

**Soundness:** 3
**Presentation:** 3
**Contribution:** 3
**Rating:** 6
**Confidence:** 4

**Summary:**

This paper aims to address three main limitations of the DPO algorithm identified by recent works: (1) preference signals only on outcomes are too coarse-grained, (2) decreasing the rewards for both preferred and nonpreferred outputs, and (3) offline. The paper proposes a new preference data collection method and a new variant of the DPO algorithm to address this. For the data collection, given an instruction, the paper uses two models (before and after SFT) to collect three pairs of outputs, and iteratively uses them in the preference learning stage. For the preference learning algorithm, the paper proposes to augment DPO with token-level uncertainties that aims to down-weight the gradient of the uncertain tokens.

Experiments on text2sql, code generation, and math reasoning are conducted with Mistral-7B and Llama3-8B. Promising results are achieved compared to DPO and SFT baselines.

I will seriously consider revising my current negative score if the authors can help me understand some technical details in the response (details below).

**Strengths:**

- The proposed data collection method and the IUPO algorithm are interesting and novel to the best of my knowledge
- Motivation is clear
- Comprehensive and interesting analysis

**Weaknesses:**

- Some key technical details are missing, which makes it difficult for me to understand the algorithm and evaluate it. Specifically, the $\odot$ operator in Eqs 4 and 5 are never explained. My guess from the context is that it denotes the elementwise product between two vectors; however, $\pi_{theta}$ should be a scalar (probability) and it is unclear to me how this works. Maybe the authors can help me understand this
- Can the authors comment on how the proposed algorithm relates to https://arxiv.org/abs/2404.12358 and https://arxiv.org/abs/2406.06887, and whether or not the paper should compare to them?
- It would be interesting to explore whether DPO can benefit from the 3-iteration training
- Probably due to LLM revision, some of the wording choices read strange and inaccurate to me. For example, "unfortunate instances" (line 150), the "precision and clarity" of the preference signals (line 138), "coincides with" (line238). I strongly recommend a thorough proof read.

**Questions:**

Can one go beyond 3 iterations?

---

> ### Author Response · Authors · 2024-11-25
> **Response to Reviewer yKKv (Part 1)**
>
> Dear Reviewer yKKv,
>
> Thank you for your valuable comments and thoughtful questions. We sincerely appreciate the time and effort you have invested in reviewing our work. We have implemented several key revisions in the draft in response to your and other reviewers’ constructive feedback. Below, we provide detailed responses to each of your comments to the best of our ability.
>
> **Response to Weakness:**
>
> **W1**. We are sorry for the confusion of our method. We have updated the equations in Section 3.3 in the revised draft version. We define the uncertainty measure \delta of an answer sequence $y$ as $\Delta(y) = \{\Delta(y^t)|t=1,...,|y|\}$, and $\Delta(y^t)$ refers to the uncertainty of token $t$ of $y$, defined as $\Delta(y^t) = p(y^t_1|y^{<t}, x) - p(y^t_2|y^{<t}, x) + \epsilon,\quad \Delta_t \in [\epsilon, 1]$. Then we revised probability of  $\pi_\theta$ as follows: $\pi_{\theta,\Delta}(y|x)=\prod_t^{|y|}\pi_\theta(y^t|y^{<t},x)\cdot\Delta(y^t)$
>
> **W2**. Thank you for your insightful questions and suggestions. Allow me to describe our approach with these two pieces of work:
>
> - [1]: This work provides a theoretical extended analysis for DPO, and demonstrates that it is possible to reframe DPO in the token-level Markov Decision Process (MDP) which enables the credit assignment ability of the method. This work also studies the well-know issue of DPO training that the implicit chosen rewards decline during the raining process. However, this paper does not give a solution to alleviate this issue. Furthermore, although DPO can assign credit to the tokens in sequence, it is still affected by the token uncertainty of the model, especially in scenarios where chosen and rejected examples are minimal contrastive.
>
> - PLUM [2]. PLUM proposes an on-policy preference learning framework that iterative optimizes the policy model via response sampling, which is similar to ours. However, the motivation and focus are different between PLUM and us. The main contribution of PLUM is the **test cases automatic generation framework** for code tasks to verify the generated responses and create preference data. The subsequent optimization algorithms used DPO and KTO with no changes. In contrast, we collect the preference data iterative sampling and execution feedback and **consider the learning state of the policy model**. In the code reasoning tasks, we use the data with test cases that do not require additional generation. We find the preference pairs collected in an iterative manner exhibit minimal contrast, and we propose the uncertainty-based preference optimization method to achieve fine-grained control and alleviate the failures of the DPO method as described in our paper.
>
> **W3**. Thank you for your valuable suggestions. We have updated the comparison experiments between our IUPO, DPO, and KTO  in Appendix C.1. The results can be seen below:
> |      |      |       |  | |        |         | | | |
> |------|----------------------|------------------|---------|---------|-----------|----------|-----------|-----------|-----------|
> | Model| Phase   | SQL   | BIRD  | Human Eval   | MBPP    | GSM8K     | MATH      | Avg.     |
> | Llama3-8B | Base    | 9.5   | 32.9  | 59.2   | 53.3    | 51.0      | 21.2      | 37.8      |
> | Llama3-8B  | SFT     | 50.0  | 52.7  | 40.9   | 57.6    | 82.5      | 43.5      | 54.5      |
> | Llama3-8B| DPO-1   | 50.8  | 52.5  | 38.4   | 55.3    | 82.6      | 43.5      | 53.9      |
> | Llama3-8B| DPO-2   | 48.3  | 54.3  | 37.2   | 55.3    | 82.7      | 43.3      | 53.5      |
> | Llama3-8B | DPO-3   | 49.1  | 53.0  | 37.2   | 55.6    | 83.1      | 43.2      | 53.5      |
> | Llama3-8B| KTO-1   | 49.1  | 53.0  | 36.6   | 58.8    | 82.6      | 43.1      | 53.9      |
> | Llama3-8B | KTO-2   | 48.2  | 53.7  | 37.2   | 57.6    | 82.6      | 42.6      | 53.7      |
> | Llama3-8B| KTO-3   | 52.6  | 53.7  | 37.4   | 58.4    | 83.5      | 43.4      | 54.8      |
> | Llama3-8B | IUPO-1  | 51.7  | 54.2  | 47.6   | 56.0    | 83.2      | 43.9      | 56.1      |
> | Llama3-8B| IUPO-2  | 52.6  | 54.6  | 48.8   | 58.8    | 83.5      | 43.8      | 57.0      |
> | Llama3-8B| IUPO-3  | 52.6  | 56.1 | 49.0 | 59.1| 83.8 | 43.9 | 57.4 |
>
> We find that KTO outperforms the vanilla DPO method and gains performance from iteration but underperforms our IUPO method. We think that our approach is more suitable for these scenarios where the preferred and dispreferred examples have a small edit distance.
>
> **W4**. Thanks for your careful reading and helpful reminders. We have fixed the typos you mentioned in the revised draft.
>
>
> We highly appreciate your valuable input, and we are committed to integrating these insights into our upcoming experiments and the refined version of our work.
>
> We hope these updates adequately address your concerns and kindly encourage you to reconsider our review score in light of these clarifications. Your contribution significantly enhances the quality of our research, and we thank you for your thoughtful feedback.

---

> ### Author Response · Authors · 2024-11-25
> **Response to Reviewer yKKv (Part 2)**
>
> **Response to Questions:**
>
> Q1. Iterations can be more than 3.  We conduct the analysis experiments of the iterations and model performance in Figure 7. We find that an increase in the number of iterations gives diminishing improvements to the model. We selected 3 iterations because they can balance performance gain and computation well. Furthermore, other works[3][4] also come to similar conclusions as well as the choice of iteration numbers.
>
> **References**:
>
> [1] From r to Q∗: Your Language Model is Secretly a Q-Function
>
> [2] PLUM: Improving Code LMs with Execution-Guided On-Policy Preference Learning Driven By Synthetic Test Cases
>
> [3] Self-Rewarding Language Models.
>
> [4] Iterative Reasoning Preference Optimization.

---

> > ### Comment · Reviewer_yKKv · 2024-11-26
> >
> > Thanks for the response, which addressed my concerns. I have increased my score.

---

> > > ### Author Response · Authors · 2024-11-28
> > >
> > > We sincerely appreciate your thorough review and the time you took to consider our responses and revisions! We appreciate the reviewer's insightful feedback, which contributes to the robustness and clarity of our research.

---

### Official Review · Reviewer_mQmU · 2024-11-04

**Soundness:** 3
**Presentation:** 3
**Contribution:** 2
**Rating:** 6
**Confidence:** 2

**Summary:**

This paper introduce an strategy for collecting preference pairs data through iterative sampling and execution feedback and a variant of the DPO(Direct Preference Optimization) algorithm named IUPO(Iterative Uncertainty-based Preference Optimization).
The authors claim that the IUPO method achieves fine-grained preference control by assessing model confidence and alleviates the distribution shift problem in offline DPO.
Moreover, the authors conduct experiments across three reasoning tasks to demonstrate the effectiveness and generalization of IUPO.

**Strengths:**

1. IUPO method is interesting and the "Formal analysis" part clearly demonstrates how the IUPO improves the DPO through uncertainty.
2. The proposed method achieves a good performance on reasoning tasks in text-to-SQL, code and mathematical, offering an overall improvement of 3.6% over the standard DPO method.
3. The motivation is clear, the analysis is coherent and well-reasoned, and the logic is sound.

**Weaknesses:**

1. Comparing IUPO only with DPO and DPOP may seem insufficient. Could the author compare the proposed algorithm with more relevant methods, such as KTO?
2. The article does not mention anything related to training cost. I would like to know how much the training cost for three iterations increases compared to a single iteration? Does the IUPO have the advantage in training cost compared to other methods?
3. There are some minor issues in the appendix: 1) In the caption of Table 7 and Table 8, the meanings of "tick" and "cross" have been reversed  2) On line 907, the format of the equal sign "=" is inconsistent with other lines of equal signs.

**Questions:**

See "Weakness"

---

> ### Author Response · Authors · 2024-11-25
> **Response to Reviewer mQmU**
>
> Dear Reviewer mQmU,
>
> Thank you for your valuable comments and thoughtful questions. We sincerely appreciate the time and effort you have invested in reviewing our work. We have implemented several key revisions in the draft in response to your and other reviewers’ constructive feedback. Below, we provide detailed responses to each of your comments to the best of our ability.
>
> **Response to Weakness:**
>
> **W1**. Thank you for your suggestions. We have updated the comparison experiments between our IUPO, DPO, and KTO in Appendix C.1. The results can be seen below:
> |      |      |       |  | |        |         | | | |
> |------|----------------------|------------------|---------|---------|-----------|----------|-----------|-----------|-----------|
> | Model| Phase   | SQL   | BIRD  | Human Eval   | MBPP    | GSM8K     | MATH      | Avg.     |
> | Llama3-8B | Base    | 9.5   | 32.9  | 59.2   | 53.3    | 51.0      | 21.2      | 37.8      |
> | Llama3-8B  | SFT     | 50.0  | 52.7  | 40.9   | 57.6    | 82.5      | 43.5      | 54.5      |
> | Llama3-8B| DPO-1   | 50.8  | 52.5  | 38.4   | 55.3    | 82.6      | 43.5      | 53.9      |
> | Llama3-8B| DPO-2   | 48.3  | 54.3  | 37.2   | 55.3    | 82.7      | 43.3      | 53.5      |
> | Llama3-8B | DPO-3   | 49.1  | 53.0  | 37.2   | 55.6    | 83.1      | 43.2      | 53.5      |
> | Llama3-8B| KTO-1   | 49.1  | 53.0  | 36.6   | 58.8    | 82.6      | 43.1      | 53.9      |
> | Llama3-8B | KTO-2   | 48.2  | 53.7  | 37.2   | 57.6    | 82.6      | 42.6      | 53.7      |
> | Llama3-8B| KTO-3   | 52.6  | 53.7  | 37.4   | 58.4    | 83.5      | 43.4      | 54.8      |
> | Llama3-8B | IUPO-1  | 51.7  | 54.2  | 47.6   | 56.0    | 83.2      | 43.9      | 56.1      |
> | Llama3-8B| IUPO-2  | 52.6  | 54.6  | 48.8   | 58.8    | 83.5      | 43.8      | 57.0      |
> | Llama3-8B| IUPO-3  | 52.6  | 56.1 | 49.0 | 59.1| 83.8 | 43.9 | 57.4 |
>
> We find that KTO outperforms the vanilla DPO method and gains performance from iteration but underperforms our IUPO method. We think that our approach is more suitable for these scenarios where the preferred and dispreferred examples have a small edit distance.
>
> **W2**. We acknowledge that the training cost increases as the iterations increase since it brings more training data and training time. However, the preference data comes from the model itself and we do not introduce external information, thus the performance gains are small but meaningful. Moreover, our method still outperforms other methods with the same resource consumption, as shown in Table 4, where more iterations with updating data outperform one iteration with more data.
>
> **W3**. Thanks for your careful reading and helpful reminders. We have fixed the typos you mentioned in the revised draft.
>
> We highly appreciate your valuable input, and we are committed to integrating these insights into our upcoming experiments and the refined version of our work.
>
> We hope these updates adequately address your concerns and kindly encourage you to reconsider our review score in light of these clarifications. Your contribution significantly enhances the quality of our research, and we thank you for your thoughtful feedback.

---

> > ### Comment · Reviewer_mQmU · 2024-11-27
> >
> > Thanks for your response.  After reading your rebuttal, I prefer to keep my score.

---

> > > ### Author Response · Authors · 2024-11-28
> > >
> > > We value your continued engagement. If you have any further questions or require additional clarification, please feel free to reach out.

---

### Official Review · Reviewer_oETz · 2024-11-05

**Soundness:** 3
**Presentation:** 3
**Contribution:** 2
**Rating:** 5
**Confidence:** 3

**Summary:**

The paper introduces Iterative Uncertainty-based Preference Optimization (IUPO) as an enhancement over Direct Preference Optimization (DPO) for large language models in complex reasoning tasks. IUPO incorporates iterative response sampling and execution feedback, specifically optimizing policy with token-level uncertainty measures. The experiments demonstrate improvements over DPO, showing IUPO’s benefits in reasoning tasks across models and datasets.

**Strengths:**

1. IUPO represents an advancement by using uncertainty measures for token-level adjustments, potentially enhancing fine-grained optimization.
2. The results demonstrate IUPO's effectiveness in improving reasoning abilities, specifically surpassing DPO on various benchmarks.
3. The paper includes a range of experimental analyses, including model confidence evaluations and the impact of iterations, which help illustrate IUPO's capabilities and limitations.

**Weaknesses:**

1. The enhancement over DPO, while valuable, is relatively modest, especially on hard tasks like math. It appears incremental given that DPO's limitations are well-documented, and similar iterative techniques exist in the literature [1].
2. Some contributions, particularly the automatic generation of preference data, lack clarity in their novelty relative to existing data generation techniques.
3. The data collection settings required the final answer is auto-verifiable by either comparing to the gold answer or code execution. In such case, a clear and robust reward is already available and it is not clear why DPO is still necessary here.

[1] Iterative Reasoning Preference Optimization. Richard Yuanzhe Pang et al.

**Questions:**

1. Continue on the weakness 3: given that your settings required to have an online automatic verifier to provide reward signal, you are able to directly apply policy gradient here. Without the need of training a reward model, the policy gradient algorithm is not much harder than DPO but have much better performance. One thing I am kinda confused by all the DPO papers nowadays is why they always get rid of the discussion with policy gradient method, especially in this kind of settings without the need of a reward model?
2. Can you explain Figure 4 step 2? What's the meaning of circle N and circle 1? If that means the number of iteration, why you are sampling from the N-th iteration and 1-st iteration for the well-learned type?

---

> ### Author Response · Authors · 2024-11-25
> **Response to Reviewer oETz**
>
> Dear Reviewer oETz,
>
> Thank you for your valuable comments and thoughtful questions. We sincerely appreciate the time and effort you have invested in reviewing our work. We have implemented several key revisions in the draft in response to your and other reviewers’ constructive feedback. Below, we provide detailed responses to each of your comments to the best of our ability.
>
> **Response to Weakness:**
>
> W1 & W2. **The motivation of preference data collection approach**. [1] and [2] propose a vanilla iterative preference optimization method via rewards computing and preference pairs selection. We also follow an iterative paradigm but with many improvements and differences in the implementation. **Firstly**, our method considers the learning state of the models by comparing the sampling responses of the naive model and the policy model. The results in Figure 9 demonstrate the effectiveness of this approach. **Secondly**, [1] and [2] apply a reward model to give rewards to the model’s responses while we simulate a virtual environment to execute synthetic responses in code and mathematical reasoning tasks.
>
> **The motivation of uncertainty-based preference optimization method**. In fact, [1] uses the vanilla DPO method to optimize the policy while [2] found it yields moderate gains and even decreases the performance on reasoning tasks and then adds a negative log-likelihood term in DPO. In contrast, our IUPO method is self-constrained with our preferred data collection method, as well as considering the failures of DPOs in complex reasoning tasks. As shown in Table 1, the preference pairs collected in the iterative manner exhibit minimal contrast (low edit distance). However, the vanilla DPO is not applicable to this scenario since it may decrease the probabilities of both the undesirable and preferred [3][4]. Therefore, we propose the uncertainty-based preference optimization method to achieve fine-grained control and demonstrate it can alleviate the DPO issue in formal analysis and experimental results.
>
> W3 & Q1. Thank you for your insightful questions. In my opinion, the iterative direct optimization method has several advantages over than policy gradient optimization method in some instances. One of the keys is direct optimization is more efficient and stable than the policy gradient method such as PPO, especially when the policy model has huge parameters. For example, the llama3 herd of models [5] also conducts iterative DPO training with rejection sampling, and they find that DPO requires less computation for large-scale models and performs better than PPO. [6] compares the online DPO and on-policy RLHF method and finds online DPO is more preferred than the method. Overall,  although PPO and other policy gradient methods may theoretically be able to obtain better results, iterative/online DPO method training is efficient and the process is stable.
>
> **Response to Questions:**
>
> Q2. We are sorry for the confusion. Some of the icons in Figure 4 correspond to Figure 2. Circle 1 and circle N refer to the first and N-th responses generated in Figure 2 “Response Sampling” step. They are sampled in the same iteration, one is preferred and another is dis-preferred. We are sorry for the confusion, and we have refined Figure 2 and Figure 4 in the revised draft.
>
> **References**:
>
> [1] Self-Rewarding Language Models.
>
> [2] Iterative Reasoning Preference Optimization.
>
> [3] Smaug: Fixing failure modes of preference optimisation with dpo-positive.
>
> [4] Towards Analyzing and Understanding the Limitations of DPO: A Theoretical Perspective.
>
> [5] The Llama3 Herd of Models.
>
> [6] Direct Language Model Alignment from Online AI Feedback

---

> > ### Author Response · Authors · 2024-11-28
> > **Kindly Request for Re-review and Discussion**
> >
> > Dear Reviewer oETz,
> >
> > As the discussion period draws to a close very soon, we sincerely appreciate your time and expertise! We kindly ask if our responses and revisions have addressed your concerns. If so, we respectfully request you to reconsider your assessment. Further, we are happy to address any further concerns you may have! Your valuable insights contribute significantly to the refinement of our work.
> >
> >
> > Best
> >
> > The Authors

---

### Official Review · Reviewer_9kF7 · 2024-11-05

**Soundness:** 3
**Presentation:** 3
**Contribution:** 3
**Rating:** 6
**Confidence:** 4

**Summary:**

The paper proposes IUPO: an iterative method for preference optimization in LLMs. The method works by iteratively (i) collecting preference data through response sampling and execution feedback from a virtual environment and (ii) optimizing the LLM with a modification of DPO that integrates an "uncertainty score". The approach has been tested on SQL, code, and math tasks and shows an overall 2.1/3.6% improvement over standard preference optimization approaches.

**Strengths:**

- The experiments are thorough and well-executed
- The approach has an overall 2.1/3.6% improvement over standard preference optimization approaches on SQL, code, and math tasks.
- The paper is well-structured and well-written:
	- The algorithm proposed is simple and clear
	- The author proposed a nice overview of the current limitations of DPO
- The code is released with the paper; it's well-structured and documented.

**Weaknesses:**

- (Major) The uncertainty definition used is not mathematically grounded and lacks a rigorous definition and connection with common uncertainty estimators; it seems more like an ad-hoc heuristic used to boost performance than a proper estimator.
- (Major) The data-collection procedure for IUPO requires Execution Feedback from a virtual environment. This drastically limits the applicability of the method to tasks with an execution environment available (e.g., coding, SQL). The generalizability of the approach to most tasks that do not have this virtual environment available is unclear.
- (Major) The proposed method does not introduce exceptional variations compared to DPO. The method just introduces (i) a different way of sampling data and (ii) a minor modification of the DPO loss with the "uncertainty score" computed
- (Medium) The paper is well-written, but the main section of your method (Section 3.3) is not very clear (see also Questions section):
	- The definition of uncertainty $\Delta_t$ is not clear (see Questions)
	- The section uses inconsistent mathematical notation that makes it hard to follow. For example the subscript in  $\Delta_t$ is used to indicate the token in the sequence but in the following equations the functional form $\Delta(\cdot)$ is used without explaining how the uncertainty for the full sequence is computed.
	- Equation 3 is quite cryptic and several details are left to the reader (e.g., the definition of relative distance $k$ and window size $K$)
	- The "formal analysis" (Equations 6-7) is almost entirely derived from [Pal et al. (2024) Equations 4-6](https://arxiv.org/pdf/2402.13228#page=20.55), however I believe there are several missing pieces in the explanation that make it hard to follow in your paper
	- It is not clear how IUPO addresses every issue raised in 3.2
- (Medium) I believe the experimental part is missing an ablation of DPO with the uncertainty-based preference optimization (without the Iterative data collection part). It's not clear the role of the uncertainty score if used in standard DPO training.

**Questions:**

- Why did you choose Equation 2 to model uncertainty instead of other approaches? The choice of subtracting the top two tokens seems quite arbitrary and not grounded. I got that Wang and Zhou inspire it, but that paper has not been peer-reviewed. Can you elaborate a little bit on this choice?
- Why do you call $\Delta_t$ "uncertainty"? If the second token is lower it means that the model is less uncertain, but $\Delta_t$ is actually higher (the opposite of uncertainty).
- "Specifically, we mine tokens with uncertainty measure below a fixed threshold τ and adjust the confidence of tokens within their subsequent window K:" + Equation 3. I did not fully get what you meant by "mine." Can you elaborate a little bit?
- "where ∆(·) is a set of uncertainty measures for all tokens in response." It is not clear to me the definition of this set. "Since ∆t is less than 1, the probability of the token after the difference with preferred in πθ(yl|x) will decrease, and the corresponding gradient will be lower, thus alleviating the decrease in the preferred probability issue." Could you elaborate?
- There are probably several typos in: Equations 4-8 $\mathcal{L}_{UPO}$ ; L323 "U-DPO" ; Figure 6 "UDPO"; Figure 5 BIRD "IU-DPO"
- Why is the weak to strong generalization experiment (Table 3) performed just on the Text-to-SQL task and without comparing it with DPO and DPOP?
- It's not clear how Figure 5 is computed. From my understanding you compute $\Delta_t$ for each token but how is this measure aggregated over the length and the dataset? This is not explained in the paper (I suppose mean). Again it is counter intuitive that $\Delta_t$  is called uncertainty while larger $\Delta_t$  means more confidence in the generation.
- "Since the parameters θ of models are numerous, we focus on the logits $\theta_j$ , which is input to softmax". What is $\theta_j$ ?
- Why did you set the threshold $\tau$ to 0.3? What is the rational behind?

---

> ### Author Response · Authors · 2024-11-25
> **Response to Reviewer 9kF7 (Part 1)**
>
> Dear Reviewer 9kF7,
>
> Thank you for your valuable comments and thoughtful questions. We sincerely appreciate the time and effort you have invested in reviewing our work. We have implemented several key revisions in the draft in response to your and other reviewers’ constructive feedback. Below, we provide detailed responses to each of your comments to the best of our ability.
>
> **Response to Weaknesses:**
>
> W1. Measuring uncertainty in language models is a popular area of research, and can be used to assess the hallucinations of models. The common uncertainty measure methods are likelihood-based, prompt-based, sampling-based, training-based, etc. However, in our paper, we focus on **token-level uncertainty**. In our early experiments, we found the size of the probability that the model **predicts a particular token positively correlates with the likelihood of making an error** in the code or math reasoning scenarios. Thus we follow [3] and [4] to define the uncertainty of the token in the generating process and alleviate the issues of DPO by refining the loss based on the uncertainty measures.
>
> W2. Our work focuses on complex reasoning tasks including mathematical and code reasoning where DPO is prone to problems and the preference pairs are minimal contrastive. In other reasoning tasks such as commonsense reasoning, we can use reward models to select the preferred and undesirable examples, and then apply the iterative framework to conduct a preference optimization process, just as other work has done [1] [2].
>
> W3. We have described the limitations of DPO in complex reasoning tasks including code and mathematical reasoning in Section 1 and Section 3.2. Compared with DPO, we made many improvements to alleviate these issues. Firstly, we introduce a cheap and efficient preference data collection approach considering the learning state of the policy model. Secondly, we iteratively optimize the policy model based on the proposed framework, making DPO online to alleviate the distribution shift problem. Last, in response to the problem of DPO decreasing the probability of preferred examples, we propose the uncertainty-based preference optimization method and theoretically and experimentally demonstrate its effectiveness.
>
> W4. We are sorry for the confusion. We have updated the following items in the revised draft version.
>
> - We define the uncertainty measure \delta of an answer sequence $y$ as $\Delta(y) = \{\Delta(y^t)|t=1,...,|y|\}$, and $\Delta(y^t)$ refers to the uncertainty of token $t$ of $y$, defined as $\Delta(y^t) = p(y^t_1|y^{<t}, x) - p(y^t_2|y^{<t}, x) + \epsilon,\quad \Delta_t \in [\epsilon, 1]$.
> - Sorry for the confusion. We have updated the equations in Section 3.
> - We select the tokens with uncertainty measure below a fixed threshold τ and then modify the uncertainty values for the subsequent K tokens. The windows size K denotes the number of subsequent tokens to be modified, and the relative distance refers to how many token gaps there are between the token in the window and the selected token.
> - In the formal analysis section, we explicitly state that we follow [5] and further extend it to our approach.
> - For issues 1 and 2, our uncertainty-based preference method enables the optimization process with fine-grained token control. And our formal analysis and experimental results demonstrate the introduction of the uncertainty measure can alleviate the preferred probability decrease issue.  For issue 3, we introduce IUPO and collect the preference data iteratively using the method proposed in Section 2, thus the IUPO method is an online method that the data and policy updated iteratively.
>
> W5. In table 2, the phrase “IUPO-1”  refers to DPO with the uncertainty-based method and without iterative training. We can demonstrate the role of the uncertainty introduce by comparing the performance between IUPO-1 and DPO.

---

> ### Author Response · Authors · 2024-11-25
> **Response to Reviewer 9kF7 (Part 2)**
>
> **Response to Questions:**
>
> **Q1**. See Re W1.
>
> **Q2**. We follow [3] to call the $\Delta(y)$ as uncertainty. The value of $\Delta(y)$ reflects the level of confidence the model has in predicting a token. As you said, the actual magnitude of this value is the opposite of what uncertainty means, but we are more concerned with uncertainty than the certainty of the models.
>
> **Q3**. If the uncertainty measure of a token is below a fixed threshold, we will adjust the confidence level of the tokens in the subsequent window, which in turn affects the calculation of the subsequent loss.
>
> **Q4**. We are sorry for the confusion. We have updated the equations in Section 3.3 in the revised draft version. The uncertainty measure \delta of an answer sequence $y$ can be defined as $\Delta(y) = \{\Delta(y^t)|t=1,...,|y|\}$, and $\Delta(y^t)$ refers to the uncertainty of token $t$ of $y$, defined as $\Delta(y^t) = p(y^t_1|y^{<t}, x) - p(y^t_2|y^{<t}, x) + \epsilon,\quad \Delta_t \in [\epsilon, 1]$.
>
> **Q5**. Thanks for your careful reading and helpful reminders. We have fixed the typos you mentioned in the revised draft.
>
> **Q6**. DPO training for the 70B model is resource intensive, due to the resource limitations, we only experimented on the Text-to-SQL task. Furthermore, we want to try whether the preference data collected with a small model iteratively would be applicable to large models, so we conducted experiments using only IUPO.
>
> **Q7**. In Figure 5, we averaged the uncertainty measure per dataset at token granularity.
>
> **Q8**. $\theta$ is the logits matrix output from language models, and \theta_j is the logits of the j-th token. Then \theta will be input to the softmax layer and we obtain the final word probabilities distribution.
>
> **Q9**. The uncertainty threshold $\tau$ and the uncertainty window K are hyper-parameters in our paper. We set the τ = 0.3 and K = 5 based on the experimental results.
>
> We highly appreciate your valuable input, and we are committed to integrating these insights into our upcoming experiments and the refined version of our work.
>
> We hope these updates adequately address your concerns and kindly encourage you to reconsider our review score in light of these clarifications. Your contribution significantly enhances the quality of our research, and we thank you for your thoughtful feedback.
>
> **References**:
>
> [1] Self-Rewarding Language Models.
>
> [2] Iterative Reasoning Preference Optimization.
>
> [3] Chain-of-thought reasoning without prompting.
>
> [4] Minimum-margin active learning.
>
> [5] Smaug: Fixing Failure Modes of Preference Optimisation with DPO-Positive

---

> > ### Comment · Reviewer_9kF7 · 2024-11-26
> >
> > I thank the authors for answering my comments. I acknowledge this was no easy task given the restricted time and number of reviews received.
> >
> > > we found the size of the probability that the model **predicts a particular token positively correlates with the likelihood of making an error**
> >
> > I understood this aspect. What I was highlighting in the original review as a major weakness was the lack of a grounded and rigorous definition of the metric used. As you said, there are several methods to estimate token-level uncertainty in the literature (see [here](https://aclanthology.org/2023.emnlp-demo.41/) for example), but the method used seems more guided by intuition instead of being connected to any of the known estimators. Also, I believe that calling it uncertainty is misleading as 1) it's not a formal uncertainty estimator and 2) "the actual magnitude of this value is the opposite of what uncertainty means" as you acknowledged.
> >
> > > W2. Our work focuses on complex reasoning tasks including mathematical and code reasoning where DPO is prone to problems and the preference pairs are minimal contrastive.
> >
> > I thank the authors for this answer. I think it might be beneficial to highlight that DPO is deficient in mathematical and reasoning tasks.
> >
> > > In other reasoning tasks such as commonsense reasoning, we can use reward models to select the preferred and undesirable examples, and then apply the iterative framework to conduct a preference optimization process, just as other work has done [1] [2].
> >
> > If IUPO can be used in general settings with a reward model, this raises the question of how IUPO would perform with the additional uncertainty part compared to [1-2].
> >
> > I thank the author again for their reply and for answering all my questions. However, I still believe there are significant weaknesses in the methodology that need to be addressed, particularly points W1 to 3. I strongly encourage the authors to focus on resolving these issues. As such, I feel it is appropriate to maintain my current score at this time.

---

> > > ### Author Response · Authors · 2024-11-28
> > > **Response to Reviewer 9kF7 Follow-Up**
> > >
> > > Dear Reviewer **9kF7**:
> > >
> > > Thank you for your thorough consideration and further reply. Allow us to make some further clarifications:
> > >
> > > Thank you for your valuable and professional question. However, we do not think we should connect our method with the other uncertainty estimator method for LLM. **Firstly**, the motivation of our “uncertainty measure” is different from other “uncertainty estimators”. We measure the uncertainty of the LLM on a certain token to influence the preference optimization process on the gradient of the model. Most “uncertainty estimators” are designed to get the confidence of the model at the sentence or data level to make the model more reliable. The inconsistency of purpose can make differences in specific approaches. **Secondly**, we focus only on the probability values of the token level of the model during the prediction process, and different tokens can be viewed as independently and identically distributed from each other. It is most straightforward to use the probability distribution predicted by the model directly. For example, we can use the maximum probability,  average probability, or the mapping of a probability distribution of the model’s prediction to measure confidence. This can be linked to some white-box ”uncertainty estimators“, but we don't think it's equivalent under the conditions mentioned earlier. For the specific implementation, we choose the difference between the two maximum probabilities because of other work[1] and our experimental validation. Furthermore, this approach is similar to the minimum-margin method [2], which is supervised by the model uncertainty. Overall, we think our uncertainty is different from “uncertainty estimators”, but rather a way to measure model confidence on a certainty token, and further refine the gradient.
> > >
> > > > If IUPO can be used in general settings with a reward model, this raises the question of how IUPO would perform with the additional uncertainty part compared to [1-2].
> > > >
> > >
> > > We have updated comparison experiments between our IUDPO and DPO on ARC-c and TruthfulQA datasets in Appendix C.2. Specifically, we chose a reward model to replace the virtual execution environment to verify the correctness of the responses. Responses with high reward scores are used as preferred examples and those with low scores are used as dispreferred examples. The experimental settings are the same as [3]. The results can be seen as follows:
> > >
> > > | Metric | ARC-c | TruthfulQA |
> > > | --- | --- | --- |
> > > | SFT | 60.0 | 52.0 |
> > > | DPO-1 | 63.5 | 50.0 |
> > > | DPO-2 | 64.9 | 52.5 |
> > > | DPO-3 | 66.8 | 53.8 |
> > > | IUPO-1 | 63.4 | 51.0 |
> > > | IUPO-2 | 65.2 | 53.4 |
> > > | IUPO-3 | 66.9 | 54.7 |
> > >
> > > We can see that our IUPO obtains superior performance than DPO in each iteration, and the performance improves with iterations increase, proving that our approach is also feasible with the reward model scenarios.
> > >
> > > Thank you again for your thoughtful review and constructive feedback. We hope these additional experiments and clarifications address your concerns and kindly encourage you to reconsider your score in light of the updates! We are grateful for the opportunity to engage further.
> > >
> > > [1] Chain-of-Thought Reasoning without Prompting.
> > >
> > > [2] Minimum-Margin Active Learning.
> > >
> > > [3] RLHF Workflow: From Reward Modeling to Online RLHF.

---

> > > > ### Comment · Reviewer_9kF7 · 2024-11-28
> > > >
> > > > I would like to thank the authors once again for dedicating their time and effort to addressing my comments.
> > > >
> > > > >However, we do not think we should connect our method with the other uncertainty estimator method for LLM
> > > >
> > > > I understand that it is a common practice in literature to propose ad-hoc solutions that are sound but not grounded in math. Although I personally disagree with this approach, given its prevalence, I believe it would be beneficial to at least discuss how your method relates to existing uncertainty estimators.
> > > >
> > > > > Most “uncertainty estimators” are designed to get the confidence of the model at the sentence or data level to make the model more reliable.
> > > >
> > > > This statement is not entirely accurate. There are multiple token-level uncertainty estimators in the literature. For instance, a straightforward and computationally inexpensive token-level estimator is token entropy over the vocabulary (see [here](https://aclanthology.org/2023.emnlp-demo.41/) for more estimators).
> > > >
> > > > >This can be linked to some white-box ”uncertainty estimators“, but we don't think it's equivalent under the conditions mentioned earlier.
> > > >
> > > > I appreciate your clarification, and I apologize if my previous comments caused any misunderstanding. My intention was not to force a direct link to specific white-box uncertainty estimators from the cited paper. Rather, my original concern was that your "estimator" appears more like an ad-hoc heuristic. I believe this is a major weakness that could be strengthened by grounding the approach in math with known estimators.
> > > >
> > > > > Furthermore, this approach is similar to the minimum-margin method [2], which is supervised by the model uncertainty. Overall, we think our uncertainty is different from “uncertainty estimators”, but rather a way to measure model confidence on a certainty token, and further refine the gradient.
> > > >
> > > > Thank you for elaborating on this point. However, given the potential for confusion, I suggest considering an alternative term instead of "uncertainty," which could be misleading. This adjustment would clarify your method’s scope and distinctiveness, avoiding comparison with known estimators. I believe "uncertainty" is not the right term, considering that the quantity is actually the opposite of commonly known uncertainty.
> > > >
> > > > > We have updated comparison experiments between our IUDPO and DPO on ARC-c and TruthfulQA datasets in Appendix C.2
> > > >
> > > > I appreciate the additional experiments you have provided. While IUPO shows slight improvements over DPO, the performance gain appears marginal. Nevertheless, I have slightly increased my score in light of these updates.

---

> > > > > ### Author Response · Authors · 2024-11-29
> > > > > **Thank you for your feedback**
> > > > >
> > > > > Dear Reviewer 9kF7:
> > > > >
> > > > > We sincerely appreciate your thorough review and insightful feedback. Your detailed comments and constructive feedback are invaluable to us. We will carefully incorporate your suggestions to enhance our manuscript, particularly regarding the mathematical formulation of uncertainty and the associated terminology.
> > > > >
> > > > > Best,
> > > > >
> > > > > Authors

---

### Official Review · Reviewer_6qtw · 2024-11-06

**Soundness:** 2
**Presentation:** 3
**Contribution:** 2
**Rating:** 5
**Confidence:** 4

**Summary:**

This paper introduce Iterative Uncertainty-based Preference Optimization (IUPO) to improve the reasoning abilities (math and coding specifically in this paper) of large language models. Previous DPO always fail in complex reasoning tasks which require long reasoning chains because of the scarcity of high-quality preference data and the inherent limitation including coarse-grained (response-level) preference signal and decrease in preferred probability. To tackle these challenges, the authors propose the IUPO which automatically generates preference data (for math and coding) by compare the generated results with ground truth and additionally apply uncertainty measure to improve the models' confidence. Results on 6 datasets and plenty of analysis seem to show the improvements of IUP.

**Strengths:**

1. The proposed IUPO works well for code and math settings.

2. The introduced modification of DPO based on uncertainty makes sense.

**Weaknesses:**

However, there are several limitations:
1. It seems that the answer extractor and executable environment are specifically tailored for code and math setting. As a results, can this framework be generalized to other reasoning tasks such as spatio/commonsense reasoning or even planing task?

2. Also, since the preference creation process are relied on the accuracy of the answer extractor, what is the extraction accuracy? Will the extraction quality affect IUPO a lot?

3. Are the models learned with IUPO on math and code data be generalized to other reasoning settings? The author might want to present some out-of-domain evaluations if the claim is to improve the reasoning abilities.

**Questions:**

1. How would you select the number of iterations? Are there any creteria?

2. What do you think of the computation overhead vs performance gains? It seems that 3 or more iterations might results in extensive computation overhead while the performance improvements seems to be marginal.

---

> ### Author Response · Authors · 2024-11-25
> **Response to Reviewer 6qtw**
>
> Dear Reviewer 6qtw,
>
> Thank you for your valuable comments and thoughtful questions. We sincerely appreciate the time and effort you have invested in reviewing our work. We have implemented several key revisions in the draft in response to your and other reviewers’ constructive feedback. Below, we provide detailed responses to each of your comments to the best of our ability.
>
> **Response to Weaknesses:**
>
> **W1**. We use the answer extractor or executable feedback as verifiers in code or mathematical reasoning tasks since they can guarantee the correctness of the judgment and efficiency. For other reasoning tasks such as commonsense reasoning, we can use reward models to select the preferred and undesirable examples, and then apply the iterative framework to conduct a preference optimization process.
>
> **W2**. The answer extractor is applied in the mathematical reasoning tasks. In this scenario, the responses of LLMs follow a fixed response format as shown in Table 7, i.e. the final part is “The answer is {answer}.” So the accuracy of the answer extractor can be ensured. However, in other scenarios where it uses the reward model as a verifier, the performance of the reward model affects the quality of the preference data and can have an impact on the preference optimization process.
>
> **W3**. Thanks for your insightful suggestions. However,  in our work, we focus on complex reasoning tasks including mathematic and code reasoning tasks, and the preference data only contains information about these scenarios. We do not think that models trained in this way can be generalized well to other scenarios.
>
> **Response to Questions:**
>
> **Q1**. The choice of iteration numbers comes mainly from experimental experience. In Section 4.4, we conduct some analysis experiments of the iterations such as model performance within various iterations. As shown in Figure 7, we find that an increase in the number of iterations gives diminishing improvements to the model. We selected 3 iterations because they can balance performance gain and computation well. Furthermore, other works[1][2] also come to similar conclusions as well as the choice of iteration numbers.
>
> **Q2**. We acknowledge that the training cost increases as the iterations increase since it brings more training data and training time. However, the preference data comes from the model itself and we do not introduce external information, thus the performance gains are small but meaningful. Moreover, our method still outperforms other methods with the same resource consumption, as shown in Table 4, where more iterations with updating data outperform one iteration with more data.
>
> We highly appreciate your valuable input, and we are committed to integrating these insights into our upcoming experiments and the refined version of our work.
>
> We hope these updates adequately address your concerns and kindly encourage you to reconsider our review score in light of these clarifications. Your contribution significantly enhances the quality of our research, and we thank you for your thoughtful feedback.
>
> **References**
>
> [1] Self-Rewarding Language Models.
>
> [2] Iterative Reasoning Preference Optimization.

---

> > ### Comment · Reviewer_6qtw · 2024-11-26
> >
> > Thanks for the response!  I will keep my current score!

---

> > > ### Author Response · Authors · 2024-11-28
> > > **Response To Reviewer 6qtw Follow-Up**
> > >
> > > Dear Reviewer 6qtw,
> > >
> > > Thank you for your further reply. We are very sorry that due to time and resource limitations, we are not able to add generalization experimental results to other reasoning tasks before.
> > >
> > > Thanks to the extension of the review period, we have updated comparison experiments between our IUDPO and DPO on ARC-c and TruthfulQA datasets in Appendix C.2. Specifically, we chose a reward model to replace the virtual execution environment to verify the correctness of the responses. Responses with high reward scores are used as preferred examples and those with low scores are used as dispreferred examples. The experimental settings are the same as [3]. The results can be seen as follows:
> > >
> > > | Metric | ARC-c | TruthfulQA |
> > > | --- | --- | --- |
> > > | SFT | 60.0 | 52.0 |
> > > | DPO-1 | 63.5 | 50.0 |
> > > | DPO-2 | 64.9 | 52.5 |
> > > | DPO-3 | 66.8 | 53.8 |
> > > | IUPO-1 | 63.4 | 51.0 |
> > > | IUPO-2 | 65.2 | 53.4 |
> > > | IUPO-3 | 66.9 | 54.7 |
> > >
> > > We can see that our IUPO obtains superior performance than DPO in each iteration, and the performance improves with iterations increase, proving that our approach is also feasible with the reward model scenarios.
> > >
> > > Thank you again for your thoughtful review and constructive feedback. We hope these additional experiments and clarifications address your concerns and kindly encourage you to reconsider your score in light of the updates! We are grateful for the opportunity to engage further.

---

> > > > ### Author Response · Authors · 2024-12-02
> > > > **Kindly Request for Re-review and Discussion**
> > > >
> > > > Dear Reviewer 6qtw,
> > > >
> > > > As the discussion period draws to a close very soon, we sincerely appreciate your time and expertise! We kindly ask if our new responses and revisions have addressed your concerns. If so, we respectfully request you to reconsider your assessment. Further, we are happy to address any further concerns you may have! Your valuable insights contribute significantly to the refinement of our work.
> > > >
> > > > Thank you once again for your support and collaboration.
> > > >
> > > > Best regards,
> > > >
> > > > Authors

---

### Author Response · Authors · 2024-11-25
**Summary of Revisions**

Dear Reviewers and AC:

We express our sincere gratitude for your invaluable time and thoughtful feedback.

To address your comments, we have implemented several key revisions in response to your constructive feedback. The changes are summarized below:

1. (For Reviewer **9kF7**, **yKKv**) We have revised the method steps in Section 3.3 to provide greater clarity and transparency.
2. (For Reviewer **yKKv**) We add the experimental results of DPO with 3 optimization iterations in Table 7.
3. (For Reviewer **D59P**, **mQmU**) We add the comparison results between our IUPO and KTO with 3 optimization iterations in Appendix C.1.
4. (For Reviewer **6qtw, 9kF7**) We updated comparison experiments between our IUPO and DPO on ARC-c and TruthfulQA datasets in Appendix C.2.
5. (For Reviewer **oETz**) We update Figure 4 and its captions to clarify.
6. We corrected the typos mentioned by reviewers.

We sincerely believe that these revisions address your concerns and contribute to the overall improvement of our work. We kindly encourage you to reconsider our review score in light of these updates. Your insights are invaluable, and we remain open to further discussion and refinement. Thank you for your continued consideration. We genuinely look forward to contributing our work to the ICLR community.

Thank you very much,

Authors.

---

> ### Author Response · Authors · 2024-12-03
> **Urgent Request for Re-review and Discussion**
>
> Dear Reviewers and AC,
>
> We would like to express our sincere gratitude for your invaluable time, thoughtful feedback, and constructive engagement throughout the review process. Your insights have significantly enhanced the quality of our work, and we are deeply appreciative of your efforts.
>
> As the discussion period draws to a close very soon, we would like to gently remind Reviewers 6qtw, oETz, and D59P that further discussion and a re-review of our response.  Your valuable insights contribute significantly to the refinement of our work, and we would be most appreciative of any updates to your ratings, should you feel it is appropriate.
>
> Thank you very much for your time and consideration. We look forward to hearing from you soon.
>
> Best regards,
>
> Authors

---

### Meta-Review · Area_Chair_n3Q7 · 2024-12-19

**Metareview:**

The paper presents a method named Iterative Uncertainty-based Preference Optimization (IUPO) aimed at improving the reasoning abilities of large language models (LLMs). The authors highlight the shortcomings of Direct Preference Optimization (DPO), especially its reduced effectiveness in complex reasoning tasks like mathematics and code reasoning. These limitations stem from the lack of high-quality preference data and the constraints of the alignment method employed in DPO. IUPO automates the generation of preference data, reducing the need for manual annotations by using existing model responses and feedback. Its iterative approach ensures the data stays relevant and improves performance more effectively than simply increasing data volume. However, the current work seems to struggle in achieving significant improvements on larger datasets, and whether it can be generalized to a broader range of reasoning tasks remains to be further validated by the authors.

**Additional Comments On Reviewer Discussion:**

This is a very marginal paper. Before the discussion phase, two reviewers gave positive feedback, while four reviewers gave negative feedback. During the discussion period, the authors responded carefully to the reviewers' comments, added more experiments, and provided further explanations. One reviewer changed the negative opinion to a positive one, one reviewer did not respond, and other reviewers maintained their original score. The reviewers' concerns regarding the experimental setup, clarity of the paper, and some experimental results were effectively addressed by the authors. The reviewers who have not changed their scores and still hold negative opinions have concerns about the applicability of the paper and the performance improvement.

---

### Decision · Program_Chairs · 2025-01-22

Reject